# Large-scale control of the retroflection of the Labrador Current

**Mathilde Jutras** [1] ✉, **Carolina O. Dufour** [2], **Alfonso Mucci** [1,3] &
**Lauryn C. Talbot** [2]

The Labrador Current transports cold, relatively fresh, and well-oxygenated waters within the subpolar North Atlantic and towards the eastern American continental shelf. The relative contribution of these waters to either region depends on the eastward retroflection of the Labrador Current at the Grand Banks of Newfoundland. Here, we develop a retroflection index based on the pathway of virtual Lagrangian particles and show that strong retroflection generally occurs when a large-scale circulation adjustment, related to the subpolar gyre, accelerates the Labrador Current and shifts the Gulf Stream northward, partly driven by a northward shift of the wind patterns in the western North Atlantic. Starting in 2008, a particularly strong northward shift of the Gulf Stream dominates the other drivers. A mechanistic understanding of the drivers of the Labrador Current retroflection should help predict changes in the water properties in both export regions, and anticipate their impacts on marine life and deep-water formation.

Over the last decades, the Slope Sea and northeastern American continental shelf have experienced an increase in water temperatures and a decrease in oxygen concentrations[1–3], including in connected bodies of water such as the St. Lawrence Estuary[4,5] and the Gulf of Maine[6,7], with dire consequences on marine ecosystems[8,9] and fisheries[7,10]. From 2012 to 2016, the subpolar North Atlantic experienced a strong freshening[11], with potential impacts on the Atlantic meridional overturning circulation (AMOC)[11,12]. Both the deoxygenation and temperature increase over the shelf as well as the freshening of the subpolar Atlantic, have been attributed to increased transport of Labrador Current Water within the subpolar North Atlantic, at the expense of the Slope Sea and the eastern American continental shelf[4,11].

The Labrador Current is fed by a combination of waters flowing from the West Greenland Current and exported from the Arctic along the Labrador Shelf (Fig. 1a). It is characterized by two branches: an inshore branch that flows on the Labrador Shelf and an offshore branch that flows along the Labrador shelf-break. Near the tip of the Grand Banks, the current splits: part of the current retroflects northeastward to join the North Atlantic Current (NAC), and part continues along the shelf to the west[13–20] (Fig. 1a). This area lies at the confluence of the subtropical and subpolar gyres and hence at the meeting point

between the Gulf Stream (or North Atlantic Current, NAC) and the Labrador Current. Though of key importance to the circulation and water properties of the northwestern Atlantic, the retroflection of the Labrador Current and its drivers are still poorly understood[14].

Several drivers of the retroflection of the Labrador Current have been proposed in the literature. The retroflection would be controlled by wind patterns over the Labrador Shelf[11,21,22] and by the strength of the Labrador Current[4,22–24]. Some studies have proposed that a weak Labrador Current retroflection is concurrent with a strong North Atlantic oscillation (NAO)[18,25], while others have related it to a weak NAO[22], as well as with a strong AMOC[12,26]. In addition, a strong Labrador Current retroflection was found to be concurrent with a northward shift of the Gulf Stream[2,12,27], and several studies invoked that interactions with Gulf Stream/NAC eddies and meanders could divert the Labrador Current offshore[28,29] or block the inflow of the Labrador Current toward the Scotian Shelf[30,31]. Seasonal stratification in the Grand Banks region would also affect the export of freshwater away from the shelf[14].

Here, we reconcile these different perspectives by presenting evidence that retroflection occurs in a context of large-scale adjustment of the subpolar gyre circulation, partially driven by winds and

[1]Department of Earth and Planetary Sciences, McGill University, Montreal, QC, Canada. [2]Department of Atmospheric and Oceanic Sciences, McGill University, Montreal, QC, Canada. [3]Geotop, Université du Québec à Montréal, Montreal, QC, Canada. ✉e-mail: mathilde.jutras@mail.mcgill.ca

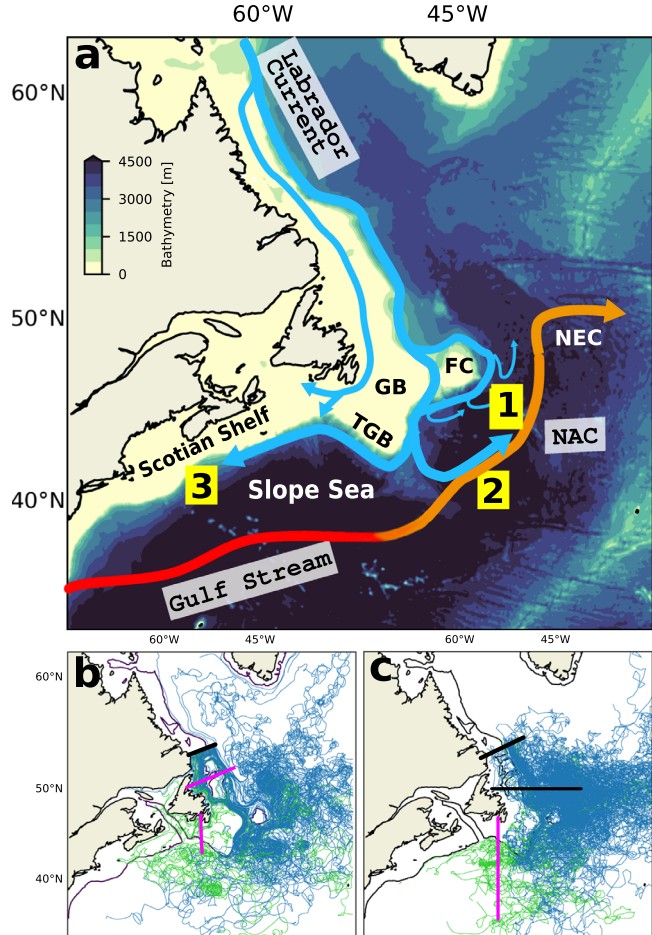

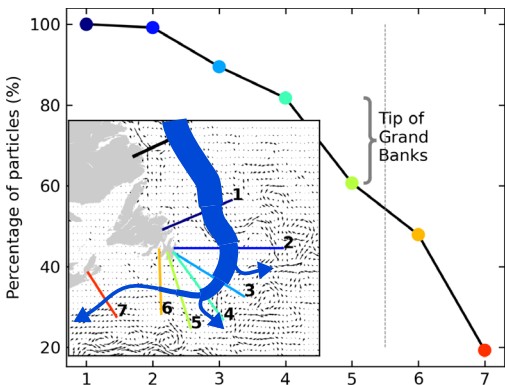

**Fig. 2 | Labrador Current retroflection location.** Percentage of the total number of particles that cross each hydrographic section (x-axis) identified in the inset for the 1994–2015 time period. Recirculating particles are counted only once. The curly brace indicates the loss at the tip of the Grand Banks. Inset: The Labrador Current is represented in blue, and its volume transport is indicated by its width. The arrows illustrate the progressive loss (i.e., leakage) of Labrador Current Waters. Beyond hydrographic section #5 (indicated by the vertical dashed line), particles are not counted as retroflected when they leak out of the Labrador Current.

## Results
### Retroflection of the Labrador Current
We examine the retroflection of the Labrador Current from Lagrangian tracking experiments where virtual particles are tracked using velocity fields of the ocean reanalysis GLORYS12V1 over the period 1993–2018 (see "Method" section). For comparison, the circulation and volume transport, as well as the retroflection, are also studied from an Eulerian perspective and presented in Supplementary Material B. The trajectories of the virtual particles reveal that, from the Grand Banks, the Labrador Current predominantly follows a seesawing system composed of two branches: a westward branch, feeding the Slope Sea and the eastern American continental shelf, that accounts for about a quarter of the Labrador Current transport downstream of the Grand Banks over 1993–2015, and an eastward branch (the retroflected branch) joining the NAC that accounts for about 60% of the transport (Supplementary Fig. S13). The retroflection occurs mostly between Flemish Cap and the tip of the Grand Banks, as well as at the tip of the Grand Banks (respectively ~25% and ~30% of the particles leaving the shelf, Fig. 2). These locations coincide very well with those observed for the Deep Western Boundary Current along the Labrador Shelf[32,33]. The pathways of the virtual particles and their relative importance are overall in good agreement with what is observed from the trajectories of surface drifters, Argo, and RAFOS/SOFAR floats (Fig. 1b, c and Supplementary Material C).

We evaluate the variability of the retroflection of the Labrador Current over 1993–2015 with an index counting retroflected virtual particles (Fig. 1b and 3; see the "Method" section). The index follows variations in temperature and salinity in the subpolar North Atlantic, in the Slope Sea, and over the northeastern American Shelf (Fig. 4a, b, correlation coefficient > 0.55, p < 0.001, Supplementary Figs. S1 and S4), further confirming the seesawing nature of the system and the influence of the Labrador Current in these regions. A strong (weak) retroflection is associated with positive (negative) salinity and temperature anomalies in the Slope Sea and along the Scotian Shelf, and to negative (positive) salinity anomalies in the subpolar North Atlantic, all the way to the eastern side of the basin (Fig. 4a, b). The freshwater input by the Labrador Current toward the subpolar North Atlantic is concentrated in the region east of the Northwest Corner, north of ~50°N, and then spreads eastward with the NAC (Fig. 4a)[11,19,34]. From the Slope Sea, the salty, warm, poorly oxygenated Gulf Stream waters penetrate adjacent channels such as the Laurentian Channel (Fig. 4a). Quantitatively, an increase in the retroflection index by 1-$\sigma$ decreases

**Fig. 1 | Labrador Current circulation. a** Schematic of the ocean circulation in the region of interest. The background color shows the bathymetry of the GLORYS12V1 model. The thick colored arrows indicate the approximate location of the main currents in the area, with NAC referring to the North Atlantic Current. In this paper, we are interested in both the shelf and shelf-break branches of the Labrador Current. Hence, we consider them together and refer to them as the Labrador Current. Numbers indicate the main pathways of the Labrador Current in the Grand Banks area, as revealed by the trajectories of the virtual particles: (1) diverted eastward between Flemish Cap and the tip of the Grand Banks, (2) diverted eastward at the southern tip of the Grand Banks, and (3) following a western route along the shelf-break. (1) and (2) represent two pathways of retroflection. The following topographic features are indicated: Grand Banks (GB), Tip of the Grand Banks (TGB), and Flemish Cap (FC). NEC refers to the Northeast Corner. **b** Examples of virtual particles trajectories. The thick black line marks the section along which the Lagrangian particles were initialized, and the pink lines the hydrographic sections used to calculate the retroflection index (see Section "Index of retroflection of the Labrador Current"). **c** Trajectories of Argo, RAFOS/SOFAR floats, and surface drifters over 2000–2018. We select floats that cross the two black lines, and classify them into retroflected or not according to whether they cross the pink vertical line. In **b**, **c**, the particles, floats and drifters classified as retroflecting appear in blue, and those classified as westward-flowing appear in green. The green trajectories found in the eastern region correspond to particles that move eastward after having initially moved westward. The thin black line delineates the 350-m isobath.

affecting the strength of the Labrador Current and the position of the Gulf Stream. To do so, we introduce a retroflection index that characterizes the magnitude of the retroflection of the Labrador Current over the past 25 years. This index allows us to examine directly the link between the observed oxygen, temperature, and salinity anomalies in the Labrador Current Water export zones and the magnitude of the retroflection of the Labrador Current, as well as to investigate the link between the retroflection and multiple possible drivers.

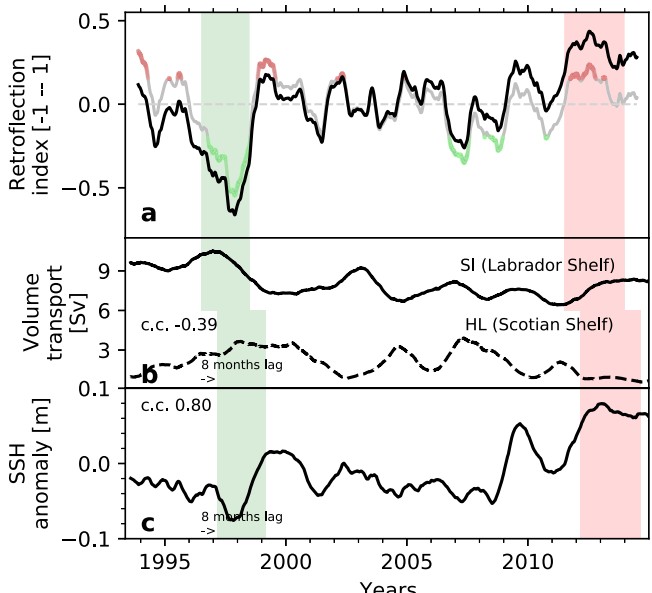

**Fig. 3 | Retroflection and forcing time series. a** Retroflection index: Total smoothed (black) and detrended (gray) indices with periods of strong (red) and weak (green) retroflection (±1σ) used for the composite analyses. Source data are provided as a Source Data file. **b** Volume transport across the SI (continuous line, on the Labrador Shelf) and HL (dashed line, on the Scotian Shelf) hydrographic sections (see Supplementary Fig. S14 for the location of the sections). **c** Sea surface height (SSH) averaged near the tip of the Grand Banks (box in Fig. 4d). The green and red shading indicates the periods of, respectively, significantly weaker and stronger retroflection that are discussed in Section "Retroflection of the Labrador Current." The periods are shifted by 8 months, which gives the best-lagged correlation and corresponds to the approximate advective time between the Labrador Shelf and the tip of the Grand Banks. c.c. denotes the correlation coefficient with the detrended non-smoothed retroflection index. The tick marks on the time axis indicate the beginning of the year. All variables are computed from the GLOR-YS12V1 reanalysis output. In the figure, seasonal variability is removed from all the variables using a 1-year running mean.

the salinity in the subpolar North Atlantic by 0.10, and increases the salinity in the Slope Sea and close to the Scotian shelf by 0.05. The additional freshwater in the North Atlantic may enhance the water column stratification and interfere with convection[35], with implications for large-scale circulation.

The retroflection index shows a strong multiannual variability, with a standard deviation of 22% over 1993–2015 (Fig. 3a). The retroflection is particularly weak from 1996 to well into 1998, in 2007, and in 2008–2009, and strong in 1994–1996, 1999, 2002, and 2011–2014. The weak 1996–1998 and strong 2011–2014 periods are particularly salient. The retroflection index exhibits a significant positive trend of +2.4% decade⁻¹, equivalent to ~10% of the inter-annual variability of the index (Fig. 3a), which exacerbates the strong retroflection period of 2011–2014. The detrended index exhibits a number of prolonged periods of weak and strong retroflection exceeding ±1-σ from the mean (highlighted in red and green, respectively, in Fig. 3a). The strong retroflection period of 2011–2014 is consistent with the results of[36]. They show an increase in the waters of Labrador Sea origin reaching the eastern North Atlantic starting in 2008, causing the intense freshening event of the subpolar North Atlantic observed over 2012–2016[11]. This period is also concurrent with temperature record highs on the eastern American continental shelf[6] and a decrease in the inflow of Labrador Current Waters into the Laurentian Channel after 2008[4] (see Supplementary Fig. S9). Our findings are also consistent with float observations that show that more Argo and RAFOS/SOFAR floats carried by the Labrador Current were retroflected in 2009 and 2012–2014 compared to other years (Supplementary Fig. S12). The

weak retroflection period of 1996–1999 is concurrent with high salinities in the subpolar North Atlantic reported over the same period[11] (Fig. 3 and Supplementary Fig. S9). Overall, field observations support the validity of our retroflection index and confirm the role of the Labrador Current variability in the 2012–2016 subpolar North Atlantic extreme freshening event[11].

## Large-scale forcing

In this section, we identify a number of forcing mechanisms that appear to play a role in controlling the magnitude of the Labrador Current retroflection. The retroflection of the Labrador Current is strongly related to the sea surface height (SSH) anomaly over the Scotian Shelf and close to the Grand Banks. Periods of strong retroflection are associated with positive SSH anomalies in these regions (Figs. 3c and 4d). These positive anomalies suggest a northward shift in the position of the Gulf Stream and of the NAC. Thus, the retroflection is stronger when the Gulf Stream/NAC is closer to the Grand Banks. An important northward shift in the position of the Gulf Stream was detected in 2008 through a change point analysis in the SSH time series at the tip of the Grand Banks[37] and coincided with a statistically significant shift in the retroflection index towards more positive phases over 2009-2015 (Fig. 3a). This northward shift of the Gulf Stream has likely played a significant role in driving the increased retroflection and subsequent anomalously high temperatures[6,37] and low oxygen concentrations[2,4] observed in the Slope Sea and on the Scotian Shelf since 2008. An increased presence of the Gulf Stream at the tip of the Grand Banks, however, does not appear to be a necessary condition for the retroflection to occur, as retroflection took place from 1994 to 1996 when the Gulf Stream was located further south (Fig. 3c).

Over most of the time series, periods of strong retroflection are associated with periods of strong Labrador Current, as shown by a convergence of the barotropic velocity streamlines in the western part of the subpolar gyre (Fig. 4f), which is indicative of an acceleration of the Labrador Current. A weak but significant positive correlation between the Labrador Current volume transport on the Labrador Shelf and the retroflection index is also found (correlation coefficient of 0.42 for the 1999–2016 period, p < 0.001; Fig. 3b and Supplementary Fig. S5). This relation suggests a role for remote, more specifically upstream, control of the retroflection of the Labrador Current. The correlation is negative when considering the volume transport downstream of the Grand Banks on the Scotian Shelf (Fig. 3b), thus confirming that as more water is diverted to the east, less feeds the Scotian Shelf current[23]. The connection between the Labrador and the Scotian shelves is further supported by significant lagged correlations of temperature and surface salinity along streams of the Labrador Current (not shown). As expected for a baroclinic flow, the volume transport and, hence, the retroflection index are correlated with the density gradient across the shelf-break at 50 m, the depth of the jet (correlation coefficient ~0.5, depending on the location along the shelf). Advection of temperature and salinity anomalies and anomalous air–sea fluxes can, therefore, indirectly influence the retroflection.

The strength of the Labrador Current was likely a predominant driver of retroflection during the 1994–1996 period when the Gulf Stream was retracted to the south, but the retroflection was not shut down (Fig. 3). Both the position of the Gulf Stream/NAC and the Labrador Current's strength are known to be related to the subpolar gyre dynamics: the NAC forms the southern limb of the gyre, whereas the Labrador Current forms its western limb. We find that, during strong retroflection, the subpolar gyre is contracted in the western basin and expanded in the eastern basin (Fig. 4f). The contraction in the western basin is associated with an acceleration of the Labrador Current as discussed earlier, while the expansion in the eastern basin allows for the propagation of the retroflected water further east (Fig. 4a).

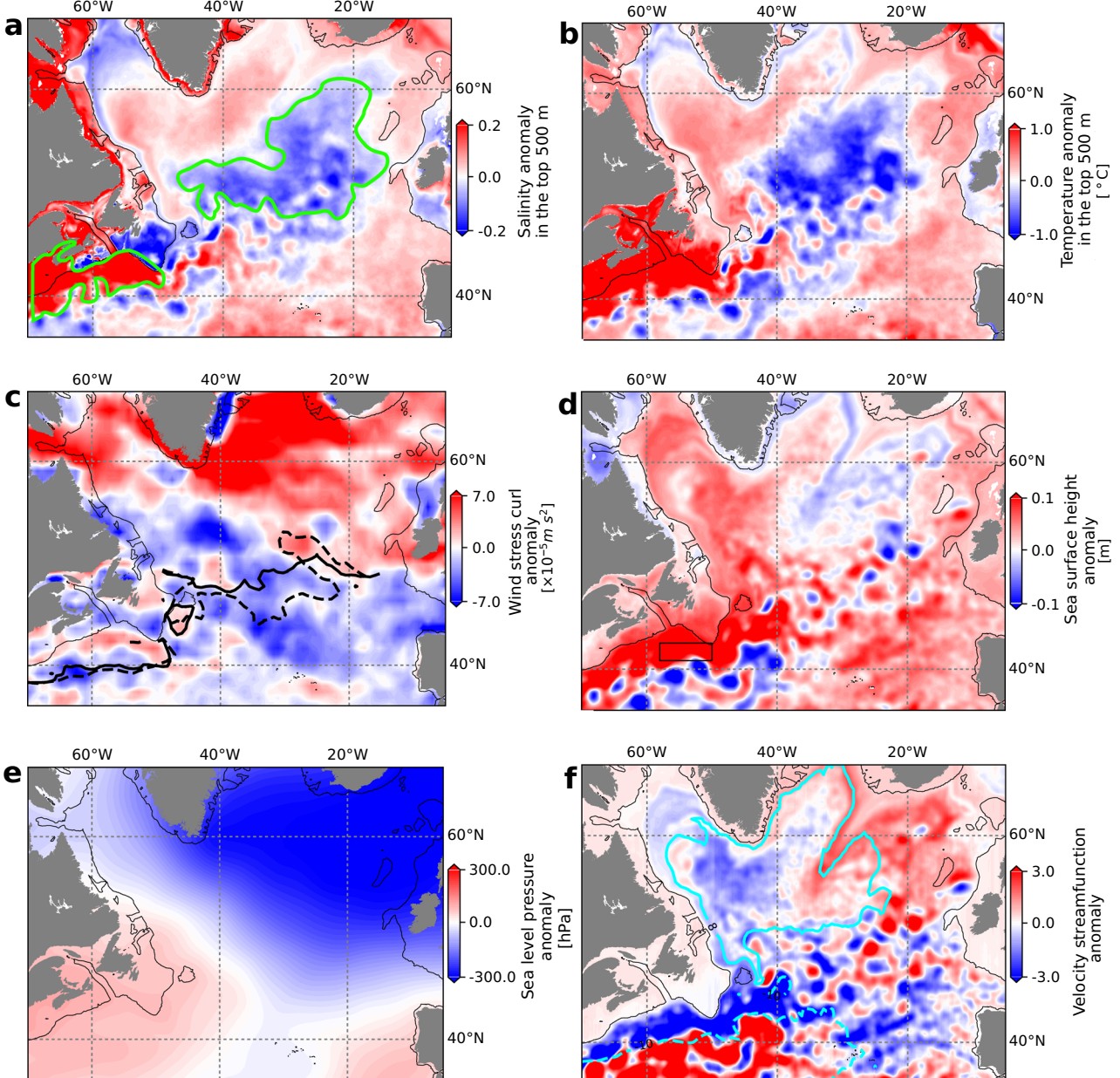

**Fig. 4 | Difference between composites of strong and weak retroflection periods (strong minus weak).** Difference between the composites for **a** average salinity over the top 500 m of the ocean, **b** average temperature over the top 500 m of the ocean, **c** wind stress curl, **d** sea-surface height, **e** sea-level pressure, and **f** barotropic stream function over the top 1 km of the ocean (see Supplementary Fig. S3 for maps of the composites). Periods used in the composites are highlighted in green (weak retroflection) and red (strong retroflection) in Fig. 3. The black line delineates the 350 m isobath. The green lines in **a** indicate regions of interest with strong differences between the composites. The dashed and full thick lines in panel **c** show the position of the lines of zero wind-stress-curl during weak and strong retroflection periods, respectively. The box in panel **d** shows the regions over which the SSH is averaged to produce the time series in Fig. 3c. It is based on the zone of strongest correlation between the retroflection index and the SSH (Supplementary Fig. S4). The cyan contours in panel **f** show climatological velocity streamlines computed over the full period (1993–2015).

The winds over the Labrador Shelf and northwestern North Atlantic also appear to influence the magnitude of the retroflection. Periods of strong retroflection are associated with negative anomalies in the wind stress curl over the Labrador Shelf and the Grand Banks (Fig. 4c). These anomalies correspond to stronger zonal winds just north of the Grand Banks that push the water offshore and to a northward shift of the line of zero wind-stress-curl in that area that promotes a contraction of the subpolar gyre. Conversely, periods of weak retroflection correspond to positive anomalies in the wind stress curl over the Labrador Shelf (Fig. 4c) and to a southward shift in the line of zero wind-stress-curl. The southward shift connects regions of positive wind stress curl located over the Labrador Sea and the Scotian Shelf (Supplementary Fig. S2), reducing the offshore push of the winds. Winds may have played a particularly important role from 1996 to 1998 when the retroflection was extremely weak, the SSH was at its lowest value over the whole time period (Fig. 3c), but the Labrador Current was strong, in contrast to the rest of the time series (Fig. 3b and Supplementary Fig. S5). During that period, the line of zero-wind-stress-curl connected the northwest North Atlantic and Slope Sea region (right panel of Supplementary Fig. S2), which could have urged the waters to move westward despite a strong Labrador Current (see Section "Discussion"). A proper evaluation of the drivers of this period

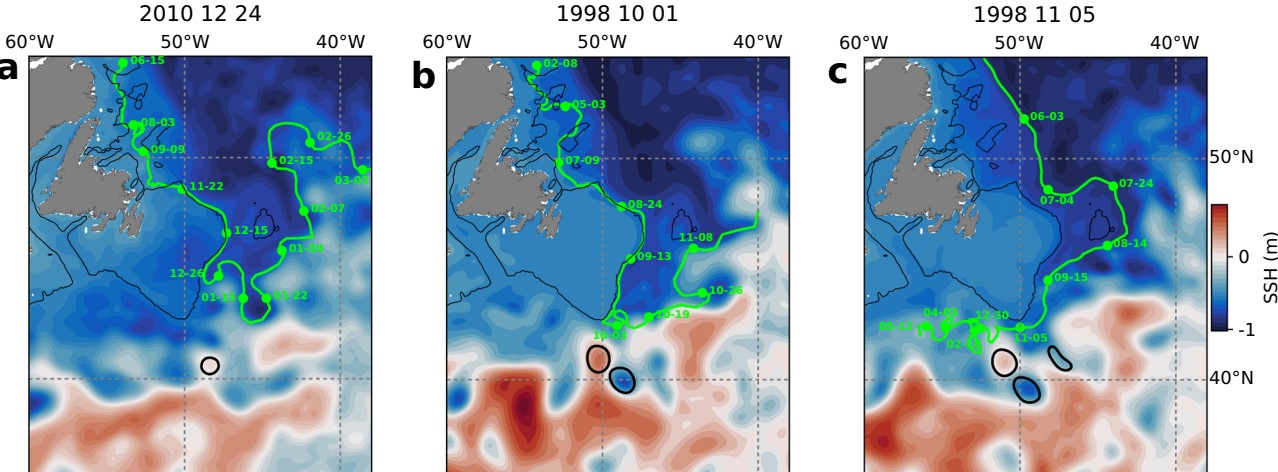

**Fig. 5 | Interactions between particles and meanders and eddies at the tip of the Grand Banks. a** Diversion of a particle by a cyclonic meander, **b** retroflection of a particle in the close presence of an anti-cyclonic eddy, **c** westward motion despite the close presence of an anti-cyclonic eddy. Thick black lines identify the cyclonic (negative sea surface height, SSH) and anti-cyclonic (positive SSH) eddies detected by an eddy detection algorithm at the tip of the Grand Banks (see "Method" section "Mechanisms controlling the retroflection"). The background colors show the sea-surface height, and the green lines show the trajectories of selected particles. The thin black lines delineate the 250 m isobath. The labels indicate the date (month–day) along the particle trajectory. The SSH field is shown for the date when the particles reach the tip of the Grand Banks (snapshot from the daily output) and are indicated in the title. All particles drift below the Ekman layer (>100 m depth).

would require a more detailed analysis and could be a subject for future studies. The relationship between the retroflection of the Labrador Current and the wind stress curl has been previously highlighted for the winter winds[11]. Shifts in the wind patterns are associated with variations in the atmospheric pressure field. During strong retroflection periods, we find that the north-south pressure difference across the jet stream is more pronounced (Fig. 4e). This pressure pattern is similar to a positive phase of the Arctic Oscillation (AO) but with a high-pressure system closer to the Grand Banks. Whereas we find significant but weak negative correlations between the retroflection index and AO and NAO indices (−0.33 and −0.26, respectively, $p < 0.0001$, no lag, Supplementary Fig. S1), we find no lagged correlation with the AMOC strength at 26°N. In summary, the enhanced meridional pressure gradient reinforces and shifts the westerly winds, pushing the Labrador Current offshore and leading to an acceleration and contraction of the western subpolar gyre. Both of these effects contribute to the retroflection of the Labrador Current.

### Retroflection at the tip of the Grand Banks

Most of the retroflection takes place at the tip of the Grand Banks (Fig. 2), where the Labrador Current and the Gulf Stream meet. The two currents are separated by a front, characterized by instabilities in the form of meanders and eddies[38,39]. At the tip of the Grand Banks, both cold cyclonic meanders and eddies generated by the tongue of the Labrador Current and warm anti-cyclonic eddies generated by the Gulf Stream are frequent (Supplementary Fig. S6). In recent years, the increased presence of the Gulf Stream at the tip of the Grand Banks concurrent with a strong retroflection of the Labrador Current has led to the hypothesis that interactions between the Labrador Current and the Gulf Stream could drive the retroflection[27,28,30,37]. Gulf Stream eddies and meanders have been shown to force the retroflection of the Labrador Current at the tail of the Grand Banks, in particular since 2008[30]. We find that there are 15% more anti-cyclonic eddies near the tip of the Grand Banks during strong retroflection events and 15% less during weak retroflection events (Supplementary Fig. S7). A higher number of anti-cyclonic eddies at the tip of the Grand Banks can be attributed to a northward migration of the Gulf Stream (Figs. 3c and 4d).

Yet, the trajectories of the particles in that area reveal that the retroflected particles are not systematically blocked or diverted by anti-cyclonic features near the tip of the Grand Banks. Anti-cyclonic features sometimes appear to block particles that are then retro-flected (Fig. 5b), but particles also retroflect even in their absence (Fig. 5a) and move westward even in their presence (Fig. 5c). Hence, whereas there typically is an increased presence of Gulf Stream eddies and meanders at the tip of the Grand Banks during strong retroflection events, these features do not appear as necessary for the retroflection of the Labrador Current to happen. Retroflected particles are mostly diverted eastward as they follow cyclonic features of circulation, especially meanders, associated with the tongue of the Labrador Current (Fig. 5a).

## Discussion

Using a retroflection index based on virtual Lagrangian particles, we evaluate the drivers of the retroflection of the Labrador Current. The position of the Gulf Stream, the Labrador Current's strength, and the westerly winds, all of which are strongly related[31] through the subpolar gyre dynamics[40], are found to play a key role in the retroflection (Section "Large-scale forcing"). Strong retroflection periods coincide with an increased meridional atmospheric pressure gradient in the subpolar North Atlantic, leading to stronger and poleward-shifted westerly winds in the western basin. Stronger westerlies push the Labrador Current offshore at the Grand Banks, directly promoting retroflection. A poleward shifted line of zero wind stress curl in the west of the basin contracts the western portion of the gyre, accelerating the Labrador Current, and shifts the Gulf Stream northward[21], indirectly promoting retroflection. The relation between a strong retroflection and a northward shift of the Gulf Stream can thus be associated with a large-scale adjustment of the circulation. In fact, whereas anti-cyclonic eddies from the Gulf Stream occasionally block the westward propagation of the Labrador Current waters near the tip of the Grand Banks, they do not appear as a necessary condition for the retroflection to happen (Section "Retroflection at the tip of the Grand Banks"). Instead, the retroflected waters follow the cyclonic meanders of the Labrador Current. Nonetheless, starting in 2008, the proximity of the Gulf Stream to the tip of the Grand Banks may have enhanced the retroflection by squeezing the Labrador Current meanders that retroflect the waters. Therefore, the position of the Gulf Stream has likely played a predominant role in driving the retroflection in recent years[28,30].

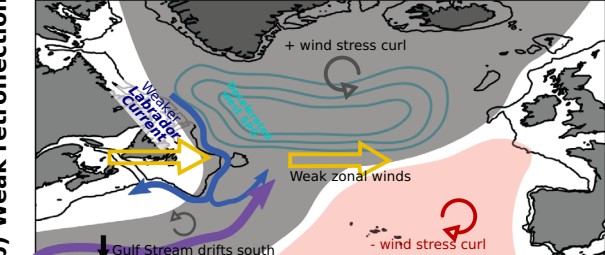

**Fig. 6 | Schematic of the oceanic and atmospheric states during strong and weak retroflection of the Labrador Current. a** During strong retroflection, negative wind stress curl anomalies over the Labrador Shelf (pink zones) reinforce zonal winds, the eastern portion of the subpolar gyre expands, its western portion contracts, the Labrador Current is accelerated, and the Gulf Stream shifts north. When the Gulf Stream comes very close to the Grand Banks, it squeezes the cyclonic Labrador Current meanders and eddies at the tip of the Grand Banks. **b** During weak retroflection, regions of positive wind stress curl anomalies (gray zones) connect over the Grand Banks area, the zonal winds are weaker, the eastern portion of the subpolar gyre contracts, its western portion expands, the Labrador Current weakens, and the Gulf Stream shifts south. The black line delineates the 350-m isobath. Part of the schematic is inspired from[11].

The role of the Labrador Current's strength in explaining the retroflection variability can be explained through a dynamical argument similar to one developed[41] for the Agulhas Current[41]. The Labrador Current is a western boundary current that hugs the coast under the Coriolis force. At the tip of the Grand Banks, the shelf edge takes an abrupt turn of more than 90° to the west (Fig. 1). Currents detach more easily from a cape when their velocity is higher[32,42]. For the Labrador Current this detachment can occur for the fast (-0.3–0.5 m s⁻¹) offshore branches of the Labrador Current, while the slower inshore branches tend to follow the continental shelf[41]. As a branch detaches from the shelf, it falls in free flow conditions and overshoots the tip of the Grand Banks toward the south due to its accumulated inertia. This southward displacement comes with a decrease in the planetary vorticity of the flow so that the relative vorticity must increase to conserve total potential vorticity (a process known as the $\beta$-compensation effect). This results in a cyclonic rotation of the flow and, hence, in a retroflection of the flow. According to this scenario, (1) a larger part of the Labrador Current would be prone to detach from the continental slope when the current is stronger, and (2) this stronger current has a higher inertia upon reaching the tip of the Grand Banks, would overshoot further south, generating a stronger $\beta$-compensation effect[41]. We, therefore, expect a tight link between the strength of the Labrador Current and its retroflection, in agreement with our results (Fig. 3). We also expect the retroflected waters to follow a cyclonic motion, as observed (Fig. 5). This mechanism can also explain the discrepancy between SSH and the retroflection index in 1994–1996 when a strong Labrador Current could have yielded a stronger retroflection despite a Gulf Stream positioned far from the Grand Banks.

To conclude, our Lagrangian analysis highlights the major role of large-scale forcing through winds and gyre dynamics in controlling the retroflection of the Labrador Current (Fig. 6), consistent with results of

previous studies that suggested such a link[4,12,21–23]. We argue that the physical blocking of the Labrador Current by the Gulf Stream at the tip of the Grand Banks[2,27,31,37] needs to be considered within the context of large-scale subpolar gyre dynamics and might have only been effective in recent years. Instead, the Labrador Current's strength and winds directly contribute to the retroflection. The fact that the wind pattern, strength of the Labrador Current, and SSH are related to the retroflection with a lag of a couple of months means that we can use these variables to monitor the export of the cold, fresh, and oxygen-rich Labrador waters towards the subpolar and coastal North Atlantic. Winds can be monitored from satellite data, while the Labrador Current strength can be monitored from the array of moorings located along the Labrador Shelf. Given the impact of the variability of the Labrador Current retroflection on the salinity, temperature, oxygen, and nutrient content in the identified export zones, this monitoring could serve to predict consequences on marine life, including fish stocks, and to set fishing quotas.

## Methods
### GLORYS12V1 ocean reanalysis
We use the global 1/12° ocean physical reanalysis GLORYS12V1[43,44] from Copernicus Marine Service (CMS). GLORYS12V1 is based on version 3.1 of the NEMO system[45] and is run with version 2 of the Louvain-la-Neuve Ice elastic-viscous-plastic sea ice Model (LIM2)[46]. Tides are not included. The model uses 50 levels on the vertical, with grid thicknesses ranging from 0.5 m at the surface to 160 m at 1000 m depth. The model is run on an Arakawa C grid at a nominal resolution of 1/12°, corresponding to ~7 km at a latitude of 45°N. The reanalysis covers the period from 1993 to 2018. It is forced with the 3 h/24 h atmospheric reanalysis ERA-Interim[47]. The bathymetry is downscaled from a resolution of 1/60° or ~1 km at 45°N in the deep ocean (ETOPO1 from NOAA) and of 1/120° or ~1 km on the coast (GEBCO-08). The assimilated data comprises 1/4° NOAA sea surface temperature (SST), altimetry-derived surface level anomaly (SLA) from CMS, in situ temperature and salinity profiles from the CMS CORAv4.1 database, and CERSAT sea-ice concentrations[44]. Observations are assimilated using a reduced-order Kalman filter with a 3-D multivariate modal decomposition of the forecast error and a 7-day assimilation cycle[48]. We use the daily outputs regrided on a centered grid.

GLORYS12V1 provides a good representation of ocean circulation, with a slight overestimation of the intensity of western boundary currents[49,50]. It reproduces the variability of the AMOC as measured at the RAPID mooring array[50]. Models with similar spatial resolutions as GLORYS12V1 have been shown to reproduce well the location and transport of the Labrador Current[51], the physics and biogeochemistry of the eastern American shelf[52], and the location of the Gulf Stream[26]. A comparison with observations shows that GLORYS12V1 represents well the location and timing of fronts and eddies, as well as the main circulation features of the Labrador Current, with an underestimation of the velocity of the Labrador shelf-break jet (Supplementary Material C). Since this study focuses on the temporal variability of the retroflection and not on its magnitude, this slight underestimation should not affect the results or our interpretations. To limit the analysis to the Labrador Current and exclude the Deep Western Boundary Current (DWBC), we only consider waters with practical salinities $S_P < 34.8$[53,54] (see Supplementary Fig. S10).

### Observational datasets
We compare the Lagrangian trajectories of virtual particles with recordings of observational instruments, namely Argo floats, RAFOS and SOFAR floats, and surface drifters. Argo floats are autonomous profilers that drift passively with ocean currents at a parking depth (typically 1000 m) and profile temperature, salinity, and pressure down to approximately 2 km every 10 days. We select the floats that cross the hydrographic line (56.7°W, 53°N)–(50°W, 54.9°N) and enter

the Grand Banks area as defined by the (55°W; 43°W)–(45°N; 50°N) box. All lines extend further offshore than those used for the virtual particles (Fig. 1b, c) to account for the fact that Argo floats typically drift deeper than the virtual particles, hence further offshore on the continental slope. This provides us with a dataset of 64 Argo floats that drifted within the Labrador Current in the proximity of the Grand Banks between 2001 and 2019.

The RAFOS and SOFAR (SOund Fixing And Ranging channel) subsurface floats are compiled from 52 experiments by the WOCE Subsurface Float Data Assembly Center. These floats drift at depths between 500 and 1000 m. The position of these floats is retrieved via acoustic methods. RAFOS floats recognize "pongs" emitted by moorings, and SOFAR floats emit "pongs" retrieved by moorings. We identify 50 drifters corresponding to the same criteria as the Argo floats between 2003 and 2007.

Surface drifters are satellite-tracked buoys deployed as part of the Global Drifter Program. The buoys drift at the surface of the ocean and are equipped with 15 m or 1 m drogues. They drift within the Ekman layer and, as such, are mostly influenced by the Ekman dynamics, unlike the floats drifting in the ocean interior. We select the drifters that move southward through a box located near the Grand Banks (55°W–41°W and 45°N–50°N). Based on these criteria, we identified 79 drifters between 2000 and 2018. To separate the floats and drifters that retroflect from those that go west, we determine if the platforms cross the 54th meridian south of the Grand Banks (pink line in Fig. 1c).

### Index of retroflection of the Labrador Current
A retroflection index is derived from Lagrangian tracking experiments of virtual passive particles. The experiments are carried out with the OceanParcels (Probably A Really Computationally Efficient Lagrangian Simulator) tool for Python[55], using the daily horizontal velocities from GLORYS12V1 and the reconstructed vertical velocities obtained by considering the non-divergence of the flow and the change in sea surface height. Virtual particles are seeded along the (53°N, 56.7°W)–(54.3°N, 52.0°W) line (Fig. 1) every 1/12° in the horizontal and every 10 m in the vertical, for a total of 966 particles per seeding event. This number is sufficient, as increasing it does not significantly alter the percentage of particles being retroflected or going westward[56–58]. Particles are released every week from 01-01-1993 to 01-01-2015 and are tracked for 3 years, with a 10-min time step. After three years, the particles have either reached the boundaries of the domain or have moved far from the Grand Banks.

Few particles circumnavigate the Grand Banks and reach the Scotian Shelf and Slope Sea, in agreement with the results of other modeling ([30] and *Myers, P., personal communication, 2021*) and float-based[19,59,60] studies, as well as with our own analysis of floats and drifters trajectories (Fig. 1c). Nonetheless, to verify whether the forward tracking experiments miss a contribution from the Labrador Current to the Slope Sea, we carried out a backtracking experiment in which particles are initialized on the Scotian Shelf and Slope Sea. The experiment confirms that less than 20% of the particles reaching the Scotian Shelf and Slope Sea originate from these regions and that the region is mostly supplied by water coming from the North Atlantic Ocean or by the outflow from the Laurentian Channel.

We define a retroflection index by first counting the number of particles passing daily through hydrographic sections located on the Labrador Shelf and on the Scotian Shelf (pink lines on Fig. 1b). The index covers the 1993–2015 period. The Lagrangian retroflection index is then computed from the difference between the number of particles crossing these two sections (Supplementary Fig. S1). The index is detrended by removing the statistically significant positive trend in the retroflection index. The smoothed index is then normalized between −1 and 1, and the mean state over the whole study period (1993–2015) is removed. For the composite analysis, the index is smoothed with a 12-month rolling average that removes high frequencies. The variability of

the index is the same for all particles, irrespective of the depth at which they were seeded.

### Mechanisms controlling the retroflection
To identify the mechanisms controlling the retroflection, we produce correlation and composite maps from the retroflection index and variables representative of the atmospheric, climatic, and oceanic states. The composite maps are computed from periods with anomalies greater than one standard deviation from the mean in the detrended smoothed retroflection index. The detrended index captures the interannual variability of the retroflection and allows us to examine the mechanisms controlling the retroflection at that time scale. We use the daily salinity, temperature, and sea surface height above geoid outputs from GLORYS12V1 and compute the daily volume transport, density, and pressure gradients from the available outputs. The wind and the sea level pressure are taken from the ERA-Interim atmospheric reanalysis, used to force GLORYS12V1. We compute the velocity stream function integrated over the top 1000 m. In addition to the variables presented in this paper, variables showing no correlation with the retroflection are discussed in Supplementary Material B. We compute different climate indices (see Section "Large-scale forcing"). The NAO index is computed from the first principal component of the sea level pressure anomaly in the region formed by (20°N,80°N)–(90°W,40°E)[61], and the AO index from the 20°N–80°N region. The AMOC transport at 26°N is computed by the CMS team.

We investigate the influence of eddies and meanders on the retroflection during weak and strong retroflection events (±1 standard deviation from the mean retroflection index). We do so by first running an eddy detection algorithm based on the Okubo–Weiss (OW) parameter at the tip of the Grand Banks (in the [55°E, 45°E], [38°N–45°N] box). Using maps of SSH anomaly at the time the virtual particles reach the tip of the Grand Banks, we then inspect whether the virtual particles interact with these eddies and with meanders (Fig. 5). The eddy detection algorithm is implemented in a Python package that follows[62,63]. The OW parameter is computed at a depth of 185 m, as it offers the best detection performance. The OW threshold is set to −0.35, and eddies smaller than 190 pixels on the 1/12° grid are not considered. For a quantitative analysis, we then compare the number of detected eddies with the retroflection index (Supplementary Fig. S7).

## Data availability
The raw model output from GLORYS12V1 used in this study are available from the Copernicus Marine Service (CMS) website, including the AMOC transport: http://marine.copernicus.eu/, product number GLOBAL_REANALYSIS_PHY_001_030, https://resources.marine.copernicus.eu/product-detail/GLOBAL_MULTIYEAR_PHY_001_030/INFORMATION, https://doi.org/10.48670/moi-00021. The ERA-interim atmospheric reanalysis used in this study are available from https://www.ecmwf.int/en/forecasts/datasets/reanalysis-datasets/era-interim. The RAFOS/SOFAR float data used in this study are available from https://www.aoml.noaa.gov/phod/float_traj/. The Argo data used in this study are available via the International Argo Program and the national programs that contribute to it (https://argo.ucsd.edu, https://www.ocean-ops.org). The Argo Program is part of the Global Ocean Observing System. The surface drifter data from the Global Drifter Program used in this study are available from https://www.aoml.noaa.gov/phod/gdp/index.php. Source data for the retroflection index are provided with this paper.

## Code availability
The Lagrangian tracking experiments were performed using the OceanParcels particle tracking tool. The OceanParcels Python package can be found at https://oceanparcels.org/. The eddy detection Python

package is available at https://github.com/jk-rieck/eddytools. The code used to perform the Lagrangian tracking analysis is provided with this paper.

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

## Acknowledgements

The Natural Sciences and Engineering Research Council of Canada (NSERC), the Fonds de recherche du Québec - Nature et technologie (FRQNT) and Ouranos funded M.J. through doctoral scholarships. This research was also funded by NSERC through Discovery grants to A.M. (grant no. RGPIN/04421-2018) and NSERC Accelerator Supplements to C.O.D. (grant no. RGPAS/2018-522502). M.J. and C.O.D. also wish to acknowledge the support of the Québec-Océan research network. The ADCP data used in these analyses were collected as part of a long-term observation program, the Atlantic Zone Monitoring Program. Many scientists have contributed to the collection, quality assurance and analyses that resulted in making these data available for research, as have the officers and crews of Fisheries and Oceans Canada's research vessels. M.J. would also like to thank N. Foukal for discussions on the subpolar gyre, J.K. Rieck for insightful ideas and for providing the eddy detection algorithm, and P. Myers for discussions on the Labrador Current and for sharing model outputs.

## Author contributions

M.J. led the development of the study, performed the analyses, produced the figures and was lead writer of the text. C.O.D. contributed to the development of the study, the interpretation of the results and the writing of the manuscript. L.T. performed the analyses of the observational dataset and produced the corresponding figures and text. A.M. and C.O.D. revised the manuscript.

## Competing interests

The authors declare no competing interests.
