## [Peer Review File · Nature Communications]

Large-scale control of the retroflection of the Labrador CurrentREVIEWER COMMENTS

Reviewer #1 (Remarks to the Author):

Dear editor and authors,

This original and well-written manuscript presents important results highlighting the main cause behind variability in the retroflexion of the Labrador Current. In particular, this study points out that the retroflexion is modulated by remote wind forcing in the Labrador Sea and may not be strongly affected by eddies along the front between the Gulf Stream and the Labrador Current. The conclusions in this manuscript are sound and well supported by the presented data analysis. The authors have provided a sufficient description of their robust methodology. Because of the importance of the Labrador Current for the thermohaline properties of the subpolar gyre and for marine life, the study has wide implications both in the field and beyond. I therefore recommend accepting this manuscript subject to minor revisions.

There are a few important questions that naturally follow from this study. The authors should either address them or list them as open research questions:

1) Line 164: Does this lack of correlation with the subtropical AMOC refer to a zero lag correlation? What about a lead-lag relationship between the retroflexion and the AMOC on a timescale of years?

2) Is the impact of the retroflexion limited only to the western subpolar gyre? What about correlations between the retroflexion and watermass transformation (or surface buoyancy fluxes) in the eastern subpolar gyre?

3) Page 8 of the Supplementary Materials, the paragraph on Eliminated Variables: What about the role of surface temperature and surface salinity along the western boundary of the Labrador Sea? Do SST and/or SSS along the western boundary of the Labrador Sea drive retroflexion anomalies even in the absence of wind-curl changes? More generally, the authors do not state very clearly whether the retroflexion is exclusively driven by wind anomalies, or whether surface density anomalies can play a role. The authors mention that surface density anomalies can affect the retroflexion indirectly via changing the speed of the Labrador Current. However, the authors do not show this and highlight only wind anomalies as drivers. If surface density anomalies cause retroflexion variability, would they also cause a contraction/expansion of the western subpolar gyre? Or is the latter a hallmark of the wind-driven retroflexion anomalies only?

Minor comments:

Line 189: "is also visible observations" "is also visible in observations"

Figure S3: "the domain is different for the sea level pressure (g-h)." The panels are not marked here. Also, the latitudes in Figure S3 are not indicated.

Figure S6: "3-year segments""3-year segments"

Reviewer #2 (Remarks to the Author):

This manuscript takes a truly comprehensive look at the circulation variability of the Labrador Current in the Grand Banks region using a data assimilating ocean model. The authors use a Lagrangian approach to try and tease apart different factors that cause the retroflexion of the Labrador Current versus its westward penetration to the Scotian and Northeast US shelf region. The effort and creativity brought to this exploration is

impressive - I have learned a lot in reading it and expect many readers of Nature Communications to appreciate this new look at the circulation.

However, additional attention to some important details should be required before publication, which I explain below.

Major:

1. I am uncomfortable with the statistical treatment of the degree of significance given with each correlation. All of the correlations with the retroflexion index are performed on variables that have been smoothed with a 12-month running mean (e.g. Line 128, Line 140, Fig 3). A smoothing operator introduces autocorrelation in a time series (in addition to any autocorrelation that already exists), reducing the degrees of freedom, which gives a false sense of the significance of any correlation. The authors should assess the role of autocorrelation in the time series in the tests of significance. There are statistical methods to jointly fit an autoregressive model and a the correlation between two variables. However, if there is no statistician on your team, a quick check on the significance can be performed with a bootstrapping or Monte Carlo approach. For example, create 1000 sets of random numbers of the same length and with the same degree of autocorrelation as your filtered time series. Correlate them with one another. You will very likely find that substantially more than 50 will be significant at the 5%. The R value corresponding with the 95th percentile of these correlations gives a sense of which correlations are significant at the 5% level given the autocorrelation of the smoothed time series.

2. I am totally confused by Figure 5. There are not enough details in Section 4.4. for me to understand what has been done to construct it, nor the text in Section 2.3 for me to interpret the findings. Here are the main questions I have:

a. Are the eddies detected by eye or with the OW algorithmic approach? Both are discussed in 4.4.

b. Are all retroflexions of particles at any depth included in the accounting, including at the surface? If so, the dynamics of the Ekman layer, which are very distinct from the interior (i.e. not even approximate conservation of potential vorticity) are getting problematically linked together. Illustrating why this could be an issue, Figure S12 shows that only 5% surface drifters and 8.5% of Argo floats can make the right hand turn at the Tail of the Grand Banks, while 25% of RAFOS/SOFAR floats do so. Surface drifters respond to wind forcing and - to a lesser degree - Argo floats can slip in the wind during profiling and surface communication.

c. How is the timing handled - i.e. where is the Lagrangian float relative to the eddy feature when the retroflexion happens?

d. Why does the Figure 5 analysis exclusively address cyclonic eddies? One of the paper's central arguments is to falsify the hypothesis of Gulf Stream eddies block the Labrador Current from making a right hand turn at the Tail of the Grand Banks. Gulf Stream eddies will be positive SSH anomalies and anticyclonic eddies, so I found the focus on cyclonic eddies totally counterintuitive. With this in mind, it is also worthwhile to revisit Supplemental Figure S14. The single snapshot of SLA when the Lagrangian particle is at the tip of the Grand Banks is insufficient to understand the role of the eddy in shaping the trajectory. An animation or set of still images like Figs 12 and 13 in Bower et al., 2011

(<https://www.sciencedirect.com/science/article/abs/pii/S0967064511000233?via=ihub>) is much more useful in understanding the circulation. As you can see in those figures, the *positive* sea level anomalies/anticyclonic eddies are the ones associated with retroflexion.

3. Depending on the answers to the previous question, I might change my view on this

comment, but I found that the overall framing of the paper's main argument presents a false choice between "local" and "remote" control of the degree of retroflexion. These two ideas do not seem mutually exclusive. It is plausible that there could be wind-driven changes to the subpolar gyre and the strength of the Labrador Current that allow the Gulf Stream and its eddies to impinge on the bathymetry more often (as suggested by the SSH-retroflexion index correlation, which is the highest correlation among all the variables). With greater proximity of the Gulf Stream to the bathymetry, there would be more likelihood for any Labrador Current particles that stray from the bathymetry to be swept to the east. With this in mind, I think it would be better to focus the framing of the paper around the positive result (remote forcing of the Labrador Current retroflexion) instead of the negative supposition (local control is secondary), which I was less convinced by given the evidence produced so far. This would mainly alter the Abstract, Section 2.3 and the Discussion.

Minor:

Abstract and L27 -It is strange to me to talk about how the Labrador Current carries the water into the subpolar North Atlantic because it originates in the subpolar North Atlantic. I'd say "transports within" rather than "carries into,"

L31 - "originating in the subarctic" be more specific about what you designate as the start of the LC.

L76 - seesaw implies one high while the other low, which I believe to be true but cannot be seen from Figure 1 trajectories

Just a note to say that I find Figure 2 extremely attractive and helpful for understanding.

Everywhere, the Neto et al, 2021 reference should be Gonçalves Neto et al., 2021 (the lead author has 2 last names).

Line 146-150; Does the dependence on wind stress vary by depth? Direct wind forcing would be likely to influence only the shallowest layers, whereas the curl could set up differences in the entire geostrophic transport

200-202 The lead sentence is repetitive with introduction. The Discussion could simply start with the sentence that is now second.

209 - The Solodoch results do not rule out Gulf Stream interactions. Particles from the leaky Labrador Current only get swept far to the east in the presence of the Gulf Stream/NAC. Plus, that paper focuses upstream of the tip of the Grand Banks.

L342 - I think the pink lines are on Fig 1b (not 1a).

Figure 1: Why is the pink line longer in 1c than 1b?

L368 - Is this the same technique as Foukal and Lozier 2017 (<https://agupubs.onlinelibrary.wiley.com/doi/full/10.1002/2017JC012798>) - if so, citation required.

Supplemental references: Jutras, Planat, and Dufour has no reference year. I assume this is the article "in prep." I am not sure how the journal handles this kind of reference.

Reviewer #3 (Remarks to the Author):

Remote control of the retroflexion of the Labrador Current by Jutras et al.

The manuscript from Jutras and colleagues highlights the important role of the Labrador Current (LC) in transporting cold, fresh and well oxygenated water not only southwestward along the western boundary but also in the subpolar North Atlantic thanks to an eastward retroflection. This role is often neglected by the literature that usually focus on the northward transport of warmer and salty water from the subtropical gyre to the subpolar gyre through the Gulf Stream and NAC. In this sense this paper is an important contribution to better understand the role of this water mass in the subpolar region.

The paper is well written and I would like to see this paper published, however I have 2 main concerns and some more specific comments that I would like the author to address before the paper is published.

My first major point regards the use of correlation to explain which forcing are more important in causing the retroflection. The authors developed an index that gives an information on how weak or strong the retroflection is. Thus, with negative values the retroflection is weak (I assume when is equal to -1 means there is no retroflection at all) and most of the water is exported southwestward along the slope Sea and with positive values the retroflection is strong (+1 I assume means all the LC turns eastward) and most of the water recirculate in the subpolar gyre moving along the NAC and few is exported southwestward. The authors give some explanations on which forcing are acting on the retroflection, whether remote or local forcing. My mainly concern is actually on the interpretation of the wind stress as one of the main remote forcing causing this retroflection. I honestly do not see from what you showed how the wind can play such an important role, and this might be because of how the authors chose to show this dependency. The authors give some correlations (Figure 3) and for example the correlation with the retroflection index and the wind stress curl is -0.28 which is in my opinion really low. However, the authors says that this is one of the main forcing. On the other hand, the SSH that has much higher correlation (0.62) falls into this local forcing which are argued in the paper are not playing an important role in the retroflection of the LC. Maybe I miss something in your argument which means it needs to be better explained. I am actually wondering if these correlations are of any meaning at all and might be misleading. Especially because as far as I understood they are calculated on a time series that refers only to some small subregion which are arbitrary chosen. And about this point I really do not see any meaning to show the wind stress anomaly time series for such a small area when the anomaly (see figure 4 b) looks really patchy. So why those subregions and not other? Does the wind stress anomaly time series in figure 3 have any meaning at all?

My second major point regards Fig. 6 which I assumed is the key figure that summarize all the results. When I look at the figure and compared with the one from Holliday et al., (2020), that inspired this schematic, I see something that do not quite fit with figure 10 from Holliday et al., (2020). During strong retroflection the authors show a contracted subpolar gyre, this is exactly the opposites of what is shown and explained in Holliday et al., (2020). During weak retroflection instead the authors show an expanded subpolar gyre which does not fit with the results and conclusions drawn by Holliday et al., (2020). I am also confused about the extension and contraction of the Subpolar gyre (SPG) as explained by the authors. As far as I know a weak gyre is also a contracted gyre and a strong gyre is an expanded gyre (see e.g. Berx & Payne, (2017); Bersch, (2002); Häkkinen & Rhines, (2004); Hátún et al., (2005)) and that is also what it is shown in Holliday et al., 2020. The authors however described exactly the opposite. Comparing figure 6 (upper panel) with figure 10f from Holliday et al., 2020, everything agrees beside the gyre that in figure 10f is expanded while in this manuscript is contracted. The opposite is for the figure 6 (lower panel) compared with figure 10d and e) from Holliday et al., 2020. I assume your conclusions come from what you observed by just looking at the zero-line of wind stress curl in figure 4b and figure S2. These figures show indeed that during strong retroflection period the zero-line of the wind stress curl is shifted northward and during weak retroflection is shifted southward, which would be an indication of a contracted gyre during strong retroflection and an expanded gyre during weak retroflection. However, the figures show only the line until 35°W and not the entire gyre. Could it be that if you have the full domain (the same domain you showed

for the Sea level pressure in figure 4d), that would actually show that in the eastern North Atlantic the zero-line shift northward during weak retroflection and southward during strong retroflection in agreement to what Holliday et al., 2020 also found? See for example their figure 9 where the wind stress curl is shown. In 2009 in the western North Atlantic the zero-line is shifted southward but on the eastern side is shifted northward. This year correspond to one of the years where you have a weak retroflection. In 2014 in their figure the zero-line is shifted northward in the western side of the North Atlantic and southward in the eastern side of the NA, this period corresponds to a period of strong retroflection in the manuscript. The fact that there is a northward versus southward shift in the eastern North Atlantic of the wind stress curl opposite to what is observed in the western north Atlantic which influences the position of the subpolar front is not something new. See for example the paper from Bersch (2002) that compares a NAO high period with a NOW low period. I recommend to include this paper into your discussion since his conclusions are exactly in line (except for this expansion/contraction of the gyre) with your conclusion. Bersch is also talking about the LC retroflection and how that influence the transport of LC either to the east along the NAC or southward along the boundary. So to conclude my suggestion is to look at the full domain of the wind stress curl (for full domain I mean the entire north Atlantic from west to east) if that is in agreement with Holliday et al., 2020 and Bersch (2002) and if not to discuss this disagreement. I would be surprise however if they disagree.

Specific comments

Line 80: I am not sure you can cite a paper that is in prep.

Line 97: Put the reference to Fig. 4a separate from the references. I thought that you were referring to Fig. 4a in Perez-Brunius et al., 2004.

Line 106: I think the time period when the retroflection is significantly weak is wrong. It should be 1996-1998. In 1999 it is already positive.

Line 106: I would also mention the other periods when the reflection is strong, not only the period 2011-2014.

Line 107: Can you observe the same freshening also in 1999-2000 and 1993-1995 since the retroflection index is also positive?

Line 112: I am confused, in your time series index the 2009 is negative (it has even green color) so mean weak retroflection. However, you mentioned that your funding are consistent with float observations which show that more Argo and RAFOS were retroflected in the period 2012-2014 (positive retroflection index) and 2009 which is negative. So, it is not consistent.

Lines 126 to 129: The authors wrote that "A strong current is generally associated with a strong retroflection as suggested by the positive correlation between the Labrador Current volume transport on the Labrador Shelf and the detrended retroflection index" which is the case for example in 2011-2014. But then in the period in 1996-1998 which is the weakest retroflection should correspond according to this correlation to a weak current. However, in Figure 3b I see the strongest transport on the Labrador Shelf. I am still wondering if these correlations and the way they are calculated are of any meaning at all. Besides, a correlation coefficient of 0.52 might be still considered as low correlated.

Line 132: The authors wrote: "The correlation is negative when considering the volume transport downstream of the Grand Banks". To be consistent a value should be given, how much negative? However, if you want to follow my suggestions, I think all these correlations might be all removed since I do not see them meaningful.

Line 140-142: The authors wrote "We find a significant anti-correlation between the retroflexion index and the state of the gyre (correlation coefficient of -0.36, $p < 0.0001$)". Is it significant because of the p value? Because again, -0.36 is not such a strong correlation, they are almost not correlated.

Lines 140-142: The authors wrote: "Since a contracted (i.e. less extended) gyre is associated with a faster circulation of its peripheral currents, this relation implies that the retroflexion is typically higher when the gyre is stronger (faster)." I do disagree with that and I do explain that in my second major comments. The authors should read these papers I have already mentioned above (Berx and Payne 2017, Hátún et al. 2005, Häkkinen and Rhines 2004, Bersch 2002).

Lines 145-147: The authors wrote: "Periods of strong retroflexion are associated with negative anomalies in the wind stress curl over the Labrador Shelf and the Grand Banks (Fig. 4b and 3d)." I do not see that from the figures, especially not from figure 3. Besides, the correlation is really low (0.28) as I mentioned in my first major point, my conclusion would be that they are uncorrelated or poorly correlated. Second, in figure 3 the only time when the wind stress curl anomaly is negative is in 2014 the rest of the time series is always positive. Thus, is it only in 2014 that the wind stress curl play a role? I want to stress here again, maybe the way these time series are calculated might be revised. Why did you make the calculation only for such a small region? Why that region and not another one? How different would be the time series if another subregion is chosen? Is it necessary to have a subregion at all? If yes why for each parameter a different subregion is chosen? How meaningful is to compare the times series from different subregions?

Lines 149-153: The authors wrote: "Conversely, periods of weak retroflexion correspond to positive anomalies in the wind stress curl over the Labrador Shelf (Fig. 4b), and to a southward shift in the line of zero wind-stress-curl. The southward shift connects regions of positive wind stress curl located over the Labrador Sea and the Scotian Shelf (Supplementary figure S2), reducing the offshore push of the winds." In light of my previous comments I suggest to revise this concept. Figure 4b only shows the zero curl west of 35°W.

Lines 162-164: Is it $p < 0.0001$? and how much is the correlation with the NAO index?

Lines 200-212: I am not sure you can exclude the local forcing and says that the retroflexion is mostly controlled remotely by the wind and the large scale circulation based on the correlations you found in the figure 3. If you base your conclusion on that figure, I would even argue that the local forcing are more important since the highest correlation is between the retroflexion and the SSH.

Lines 207-209: Yes, there is no interaction between the LC and the Gulf stream at that latitude, but there is an interaction with the LC and the NAC which is the continuation of the Gulf Stream.

Lines 215-248: See my second major comment.

Lines 245-248: I still don't understand how can you exclude the blocking effect of the Gulf Stream.

Line 251: results instead of resultts

Line 269: It is now called Copernicus marine Service (CMS)

Line 280: AVISO does not provide anymore altimetry-derived surface level anomaly (SLA). This is distributed by Copernicus Marine Service.

Lines 285-286: Could this overestimation of the WBCs affects the result of the study?

Lines 291-296: From the comparison seems like GLORYS is not able to distinguish at all between the Labrador shelf-break and the shelf jet. Is there no other comparison possible with other sections to check this? And if that is the case, how would this affect your results?

Line 310: 1000 meters instead of one kilometer.

Line 342: "(pink lines on Fig. 1a)", there are no pink lines on Fig. 1a

Line 364: The authors wrote "The AMOC transport at 26°N is computed by the CMEMS team". Beside that is CMS but shouldn't they be acknowledged?

Figures

Figure 1: I am a bit confused by figure 1b and 1c. If the blue represents the trajectories for the floats that retroreflect and in green the one that goes southwestward I would not expect any green close to the Flemish Cap and toward the east. But I can see some green at 45°W or even more eastward between 50°N and 40°N. Moreover, in Figure b and c the longitude ticks are missing. In the figure caption you wrote that you consider the shelf and shelf-break branches of the LC together and refer to them as the LC. Is that because you cannot distinguish the two branches with GLORYS?

Figure 2: I have problem to understand what the grey bar means. Usually, a bar on a plot is an uncertainty symbol. But I don't think that is the case.

Figure 3: I would consider to rethink about how the time series are calculated and about the correlations. Moreover, it is really not understandable how the SPGi is calculated and what represents. There are plenty of papers dealing with SPGi, which approach did you use? Which one you follow? This is not at all mentioned in the paper and do not see this particular index useful for your discussion.

Figure 4: Why do you only show for a b and c a smaller domain that for figure 4d when the summary in your figure 6 includes the same domain as in figure 4 d? Wouldn't make sense to have all on the same domain, the larger one and see if the zero line of the wind stress curl agrees with Hollidays et al., 2020? I repeat myself, why these subregions? Why a different subregion for each parameter?

Figure 6: See my second major comment.

Referenes

Bersch, M. (2002). North Atlantic Oscillation-induced changes of the upper layer circulation in the northern North Atlantic Ocean. *Journal of Geophysical Research*, 107(C10), 3156. <https://doi.org/10.1029/2001JC000901>

Berx, B., & Payne, M. R. (2017). The Sub-Polar Gyre Index - A community data set for application in fisheries and environment research. *Earth System Science Data*, 9(1), 259–266. <https://doi.org/10.5194/essd-9-259-2017>

Häkkinen, S., & Rhines, P. B. (2004). Decline of Subpolar North Atlantic Circulation during the 1990s. *Science*, 304(5670), 555–559. <https://doi.org/10.1126/science.1094917>

Hátún, H., Sande, A. B., Drange, H., Hansen, B., & Valdimarsson, H. (2005). Ocean science: Influence of the atlantic subpolar gyre on the thermohaline circulation. *Science*, 309(5742), 1841–1844. <https://doi.org/10.1126/science.1114777>

Authors responses to reviewers' comments

Large-scale control of the retroflection of the Labrador Current

Mathilde Jutras, Carolina O. Dufour, Alfonso Mucci, Lauryn L. Talbot

Manuscript Reference Number: NCOMMS-22-36412

Date: Jan. 6, 2023

Response to Reviewer #1

Comments

Reviewer: This original and well-written manuscript presents important results highlighting the main cause behind variability in the retroreflection of the Labrador Current. In particular, this study points out that the retroreflection is modulated by remote wind forcing in the Labrador Sea and may not be strongly affected by eddies along the front between the Gulf Stream and the Labrador Current. The conclusions in this manuscript are sound and well supported by the presented data analysis. The authors have provided a sufficient description of their robust methodology. Because of the importance of the Labrador Current for the thermohaline properties of the subpolar gyre and for marine life, the study has wide implications both in the field and beyond. I therefore recommend accepting this manuscript subject to minor revisions.

There are a few important questions that naturally follow from this study. The authors should either address them or list them as open research questions:

Authors: We greatly appreciate the reviewer's very positive and constructive comments, and provide point-by-point answers to all the comments below. Throughout the response, bold pieces of text inside citations represent text that was added or modified to the revised version of the manuscript.

Major comments

Reviewer: 1) Line 164: Does this lack of correlation with the subtropical AMOC refer to a zero lag correlation? What about a lead-lag relationship between the retroreflection and the AMOC on a timescale of years?

Authors: We also tested the correlation for lags of ± 2 years, in increments of one month. We did not find a significant correlation for any lag. We specify this in the revised manuscript: "*Whereas we find significant but weak negative correlations between the retroreflection index and AO and NAO indices (-0.33 and -0.26, respectively, $p < 0.0001$, **no lag**, supplementary figure S1), we find no **lagged** correlation with the AMOC strength at 26°N.*".

Reviewer: 2) Is the impact of the retroreflection limited only to the western subpolar gyre? What about correlations between the retroreflection and watermass transformation (or surface buoyancy fluxes) in the eastern subpolar gyre?

Authors: Given this reviewer's comment and others from reviewers #2 and 3, we extended the composite maps of Fig. 4 to cover the whole basin, including the eastern subpolar gyre. We can now see that the salinity anomalies reach far into the eastern side of the basin. We added a few words on this at L93-L96:

*“A strong (weak) retroflection is associated with positive (negative) salinity and temperature anomalies in the Slope Sea and along the Scotian Shelf, and to negative (positive) salinity anomalies in the subpolar North Atlantic, **all the way to the eastern side of the basin** (Fig. 4a,b).”*

Reviewer: 3) Page 8 of the Supplementary Materials, the paragraph on Eliminated Variables: What about the role of surface temperature and surface salinity along the western boundary of the Labrador Sea? Do SST and/or SSS along the western boundary of the Labrador Sea drive retroflection anomalies even in the absence of wind-curl changes? More generally, the authors do not state very clearly whether the retroflection is exclusively driven by wind anomalies, or whether surface density anomalies can play a role. The authors mention that surface density anomalies can affect the retroflection indirectly via changing the speed of the Labrador Current. However, the authors do not show this and highlight only wind anomalies as drivers. If surface density anomalies cause retroflection variability, would they also cause a contraction/expansion of the western subpolar gyre? Or is the latter a hallmark of the wind-driven retroflection anomalies only?

Authors:

- First, we would like to emphasize that we suggest that not only the winds, but also gyre dynamics (including the position of the Gulf Stream) and the strength of the Labrador Current play key roles in driving the retroflection of the Labrador Current. Throughout the text, modifications have been made to make sure that this point is clearly conveyed. For instance, in section 2.2 of the revised manuscript, we change the order in which the forcings are presented, putting the role of the SSH forward, hence clarifying that the wind is not the only driver of the retroflection. The abstract now reads as follows:
*“We [...] show that **strong retroflection generally occurs when a large-scale circulation adjustment, related to the subpolar gyre, accelerates the Labrador Current and shifts the Gulf Stream northward, partly driven by a northward shift of the zero-wind-stress-curl line in the western North Atlantic.**”*
The second paragraph of the discussion was modified substantially to more clearly explain how the different forcings act on the retroflection.
- Second, about the SST and SSS, there is a correlation between the retroflection index and the density gradient across the shelf-break, which is itself determined by temperature and salinity. The correlation is even stronger between the density gradient and the volume transport of the Labrador Current. This is expected, as the baroclinic velocity results from this density gradient. Hence, we had not included this finding in the original manuscript. We added this information in the revised manuscript, at L137: *“As expected for a baroclinic flow, the volume transport and, hence, the retroflection index, are correlated with the density gradient across the shelf-break at 50 m, the depth of the jet (correlation coefficient ~ 0.5 , depending on the location along the shelf).”* We also added a comment on the role of temperature and salinity on the flow and the retroflection: *“Advection of temperature and salinity anomalies and anomalous air-sea fluxes can therefore indirectly influence the retroflection.”*

- Finally, we note that the subpolar gyre, currents and surface densities are all interrelated. We highlight these interrelations at L236-L238: “*The position of the Gulf Stream, the Labrador Current’s strength and the westerly winds, all of which are strongly related (Zhang et al., 2016) through the subpolar gyre dynamics (Böning et al., 2006), [...]*”.
-

Minor comments

Reviewer: Line 189: “is also visible observations” → “is also visible in observations”

Authors: Modified as recommended.

Reviewer: Figure S3: “the domain is different for the sea level pressure (g-h).”
The panels are not marked here. Also, the latitudes in Figure S3 are not indicated.

Authors: We now extend all maps as to cover the eastern part of the domain (see reviewer 3’s comments). Labels are no longer necessary, since they are not used elsewhere. We added the latitude labels on Fig. S3.

Reviewer: Figure S6: ”3-year segments” → ”3-year segments”

Authors: We guess that the reviewer was suggesting that we change “3-years” for “3-year”.
Modified as recommended.

Response to Reviewer #2

Comments

Reviewer: This manuscript takes a truly comprehensive look at the circulation variability of the Labrador Current in the Grand Banks region using a data assimilating ocean model. The authors use a Lagrangian approach to try and tease apart different factors that cause the retroflexion of the Labrador Current versus its westward penetration to the Scotian and Northeast US shelf region. The effort and creativity brought to this exploration is impressive - I have learned a lot in reading it and expect many readers of Nature Communications to appreciate this new look at the circulation. However, additional attention to some important details should be required before publication, which I explain below.

Authors: We greatly appreciate the reviewer's very gracious and perceptive comments, and provide point-by-point answers to all the comments below. Throughout the response, bold pieces of text inside citations represent text that was added or modified to the revised version of the manuscript.

Major comments

Reviewer: 1. I am uncomfortable with the statistical treatment of the degree of significance given with each correlation. All of the correlations with the retroflexion index are performed on variables that have been smoothed with a 12-month running mean (e.g. Line 128, Line 140, Fig 3). A smoothing operator introduces autocorrelation in a time series (in addition to any autocorrelation that already exists), reducing the degrees of freedom, which gives a false sense of the significance of any correlation. The authors should assess the role of autocorrelation in the time series in the tests of significance. There are statistical methods to jointly fit an autoregressive model and a the correlation between two variables. However, if there is no statistician on your team, a quick check on the significance can be performed with a bootstrapping or Monte Carlo approach. For example, create 1000 sets of random numbers of the same length and with the same degree of autocorrelation as your filtered time series. Correlate them with one another. You will very likely find that substantially more than 50 will be significant at the 5%. The R value corresponding with the 95th percentile of these correlations gives a sense of which correlations are significant at the 5% level given the autocorrelation of the smoothed time series.

Authors: We thank the reviewer for this important comment. After consulting several statisticians, we made some modifications to our method and performed some tests, to ensure that the presented correlations are valid. First, to avoid auto-correlation errors, we performed correlations on the raw (i.e. non-smoothed) time series. For the NAO and AO indices, as well

as for the subpolar gyre area, which are computed monthly, we correlated them with monthly averages of the raw retroreflection index. Second, we performed a statistical test (the Dickey-Fuller test) on all time series to ensure that they were stationary, since non-stationarity and trends can induce false correlations between two time series. As mentioned in the original text, the retroreflection index was already detrended, and we now detrend the volume transport time series as well. We find that all the other time series are stationary. We present the updated correlation coefficients in the revised manuscript. We modified the text and figures to give the new correlation coefficients, including in the correlation maps of supplementary figure S4, where the zones with non-significant correlations increased in size, and we now represent them with hatching. We added the non-smoothed retroreflection index to Fig. S1 of the supplementary material.

To explain these adjustments, we made several modifications to the text. Part of section 4.3 now reads: *“The Lagrangian retroreflection index is then computed from the difference between the number of particles crossing these two sections (**supplementary figure S1**) and is smoothed with a 12-month rolling average that removes high frequencies (for the spectrum of the retroreflection index, see supplementary figure S12b). The index is detrended, by removing the statistically significant positive trend in the retroreflection index. The **smoothed** index is then normalized between -1 to 1, and the mean state over the whole study period (1993-2015) is removed. For the composite analysis, the index is smoothed with a 12-month rolling average that removes high frequencies.”* In section 4.4, we removed the following sentence: *“The daily time series of the investigated variables are smoothed with a 12-months rolling average.”* We also specify in Fig. 3a caption that this figure shows the smoothed time series: *“Retroreflection index: Total **smoothed** (black) and detrended (grey) indices”*.

The adjusted correlation coefficients led to the following changes:

- Most of the correlation coefficients remained similar and significant, except for two, which we detail below.
- First, the correlation with the subpolar gyre area became non-statistically-significant. We do not use this time series in the revised manuscript, following comment #2 from reviewer #3 discussing the relevance of that gyre index. In fact, the response of the retroreflection to the expansion/contraction of the subpolar gyre is different on the two sides of the basin, making a composite analysis more appropriate than the use of a single index of subpolar gyre area.
- Second, the correlation with the volume transport over the Labrador Shelf and slope is reduced. The correlation coefficient is significant (although quite low, 0.42) from 1999 to 2014. Following comment #1 by reviewer #3, we put composite maps forward instead of correlations. In the case of the Labrador Current strength, we use composites of the density of velocity streamlines. The discussion of the correlation with the Labrador Current in section 2.2 now goes as follows: *“Over most of the time series, periods of strong retroreflection are associated with periods of strong Labrador Current, as shown by a convergence of the barotropic velocity streamlines in the western part of the subpolar gyre (Fig. 4f), which is indicative of an acceleration of the Labrador Current. A weak but significant positive correlation between the Labrador Current*

volume transport on the Labrador Shelf and the retroflection index is also found (correlation coefficient of 0.42 for the 1999-2016 period, $p < 0.001$; Fig. 2b and supplementary figure S5).”

We also modified the text to discuss the new range of validity of the correlation. In 1996-1999, the retroflection is weak while the volume transport is high, in contrast with the rest of the time period. This 3-year reversal was also mentioned by reviewer #3 (referring to L126-129). The discussion of this period now reads as follows:

“Winds may have played a particularly important role from 1996 to 1998, when the retroflection was extremely weak, the SSH was at its lowest value over the whole time period (Fig. 3c), but the Labrador Current was strong, in contrast to the rest of the time series (Fig. 3b and supplementary figure S5). During that period, the line of zero-wind-stress-curl connected the northwest North Atlantic and Slope Sea region (right panel of supplementary figure S2), which could have urged the waters to move westward despite a strong Labrador Current (see Section 3). A proper evaluation of the drivers of this period would require a more detailed analysis, and could be a subject for future studies.”

- With the new calculations, the correlation with the NAO index becomes significant, albeit very low (-0.26). Hence, we modified the following sentence: *“Whereas we find a significant negative correlation between the retroflection index and AO and NAO indices (-0.33 and -0.26, respectively, $p < 0.0001$, no lag, supplementary figure S1), we find no lagged correlation with the AMOC strength at 26°N (not shown).”*

Reviewer: 2. I am totally confused by Figure 5. There are not enough details in Section 4.4. for me to understand what has been done to construct it, nor the text in Section 2.3 for me to interpret the findings. Here are the main questions I have:

Authors: Each specific comment is addressed below. Section 2.3 has been substantially modified to focus on a description of the behaviour of particles at the tip of the Grand Banks and to focus on the role of anti-cyclonic eddies in the retroflection. Some clarifications on the method were added to section 4.4, and Fig. 5 was modified to highlight specific particles that we follow along their displacement, thus better illustrating how they interact with the surrounding circulation features.

Reviewer: 2a. Are the eddies detected by eye or with the OW algorithmic approach? Both are discussed in 4.4.

Authors: We did check potential interactions between eddies and particles through visual inspection, which is why we reported it in the original manuscript. However, given that an automated detection of eddies based on the OW parameter is systematically used to investigate these interactions, we removed the mention to the visual inspection to avoid any confusion. We clarified this by changing L374 to:

“We do so by first running an eddy detection algorithm based on the Okubo-Weiss (OW) parameter at the tip of the Grand Banks (in the [55° E, 45° E], [38° N – 45° N] box). Using maps of SSH anomaly at the time the virtual particles reach

the tip of the Grand Banks, we then inspect whether the virtual particles interact with these eddies and with meanders (Fig. 5). The eddy detection algorithm is implemented in a Python package that follows Oliver et al. (2015) and Chelton et al. (2011)."

Reviewer: 2b. Are all retroreflections of particles at any depth included in the accounting, including at the surface? If so, the dynamics of the Ekman layer, which are very distinct from the interior (i.e. not even approximate conservation of potential vorticity) are getting problematically linked together. Illustrating why this could be an issue, Figure S12 shows that only 5% surface drifters and 8.5% of Argo floats can make the right hand turn at the Tail of the Grand Banks, while 25% of RAFOS/SOFAR floats do so. Surface drifters respond to wind forcing and - to a lesser degree - Argo floats can slip in the wind during profiling and surface communication.

Authors:

- First, yes, all depths were included. Below, Figure I is the same as the original Figure 5 but with different colors for virtual particles drifting at different depths. Since the virtual particles can move vertically, we consider the depth when the particles reach the Grand Banks. Because of this vertical motion, no virtual particles remain at the surface like surface floats would. We see that almost all the particles reaching the area of interest drift below the Ekman layer (<100 m), and hence are not purely wind-driven particles. This is because, for this plot, we select these particles that circumvent the tip of the Grand Banks, and hence drift at a certain depth. Figure I does not appear in the revised manuscript, as Figure 5 has been modified to address some of the other comments (see response to comments #2c,d). In the new version of the Fig. 5, we only show particles drifting below the Ekman layer. We specify this in the caption of the figure : "**All particles drift below the Ekman layer (>100 m depth)**". This question echoes a minor comment from the reviewer on whether the retroreflection is driven by different mechanisms at different depths. To address these two comments, we computed the retroreflection index for particles drifting at different depths. We mention the results of this analysis in the Method section (4.3), that the variability of the retroreflection index is the same for particles drifting at all depths: "*The variability of the index is the same for all particles, irrespective of the depth at which they were seeded.*" This result may arise as the particles can move vertically.
- Second, we agree that the surface drifters are mostly influenced by the Ekman dynamics, which differs from that in the ocean interior. We now acknowledge this difference in the observational part of the revised Method section (4.2) at L332-L334: "**They [the surface buoys] drift within the Ekman layer, and as such are mostly influenced by the Ekman dynamics, unlike the floats drifting in the ocean interior.**"
- Third, robust quantitative comparisons between the behaviour of surface drifters and RAFOS/SOFAR/Argo floats cannot be made, because the floats drift deeper than the

Figure I: Same as the original Fig. 5, but with different colors for the particles drifting at different depths.

virtual particles (in the Deep Western Boundary Current), but also because of the differences in the release locations and in the number of floats. Moreover, the number of floats in each dataset is not sufficient to offer a statistically significant comparison. We already mentioned these limitations in the supplementary material, where figure S11 is located:

“(1) Most of the Argo and RAFOS/SOFAR floats drift deeper than our virtual particles, and are thus advected by the Deep Western Boundary Current (see section Method). (2) While floats and drifters travel at a fixed depth, virtual particles can move vertically. Nevertheless, we use floats and drifters trajectories to perform a qualitative comparison with the virtual particle trajectories.” and *“Overall, there is a rather limited number of observational platforms drifting in the area and period of interest (193 in total, starting in 2000, Fig. S12). This prevents us from performing a statistical analysis of the variability of the preferential pathway of these platforms.”*

Hence, it is not possible to confirm if the difference in behaviour between the surface and deep observational platforms outlined by the reviewer is robust.

Reviewer: 2c. How is the timing handled - i.e. where is the Lagrangian float relative to the eddy feature when the retroflection happens?

Authors:

- The trajectories shown on Fig. 5 (and used for the quantitative analysis in section 2.3) are those of virtual particles that reach the tip of the Grand Banks during high/low retroflection events. The SSH fields shown on the maps correspond to that of the date when the particles reach the tip of the Grand Banks. To clarify this procedure, we modified the text in the Method section (4.4), at L390-L392, to ***“Using maps of SSH anomaly at the time the virtual particles reach the tip of the Grand Banks, we then inspect whether the virtual particles interact with these eddies and with meanders (Fig. 5).”***

The date of the eddy field is mentioned in the title of the figures. This is now indicated explicitly in the caption: ***“The SSH field is shown for the date when the particles reach the tip of the Grand Banks (snapshot from the daily output), and is indicated in the title.”***

- Following the reviewer’s comment and that from reviewer #3, Fig. 5 now shows only one trajectory on each map. This modification should better illustrate the interactions, or lack thereof, between virtual particles and eddies and meanders. We also indicate the date along the trajectory with labels, which further helps the reader to make the connection with the SSH field visible in the background. This modification should provide a better idea of the timing of the interactions or lack thereof. We explain this addition in the caption: ***“The labels indicate the date (month-day) along the particle trajectory.”***

Reviewer: 2d. Why does the Figure 5 analysis exclusively address cyclonic eddies? One of the paper’s central arguments is to falsify the hypothesis of Gulf Stream eddies block the Labrador Current from making a right hand turn at the Tail of the Grand Banks. Gulf Stream eddies will be positive SSH anomalies and

anticyclonic eddies, so I found the focus on cyclonic eddies totally counterintuitive. With this in mind, it is also worthwhile to revisit Supplemental Figure S14. The single snapshot of SLA when the Lagrangian particle is at the tip of the Grand Banks is insufficient to understand the role of the eddy in shaping the trajectory. An animation or set of still images like Figs 12 and 13 in Bower et al., 2011 (<https://www.sciencedirect.com/science/article/abs/pii/S0967064511000233?via=ihub>) is much more useful in understanding the circulation. As you can see in those figures, the *positive* sea level anomalies/anticyclonic eddies are the ones associated with retroflection.

Authors: This comment on cyclonic vs anti-cyclonic eddies raises a very good point, that we address in the revised manuscript.

- We agree that one would expect anti-cyclonic eddies to play a role in the retroflection. We had focused on cyclonic eddies because we do not observe direct interactions between particles and anti-cyclonic eddies whereby particles would systematically veer in the presence of an anticyclone, while we observe that particles often follow cyclonic meanders. However, the quantitative analysis presented in the supplementary material (Fig. S7) does show an increase in the number of anti-cyclonic eddies close to the tip of the Grand Banks during high retroflection events, and a decrease during low retroflection events. We suggest that the increase in the number of anti-cyclonic eddies is due to the northward shift in the Gulf Stream position during high retroflection periods (as revealed by the composite maps of SSH (Fig. 4d) and the positive correlation between SSH at the tip of the Grand Banks and the retroflection index (Fig. 3c)). So, to sum up, whereas there are more anticyclonic eddies at the tip of the Grand Banks during high retroflection events, these eddies do not necessarily directly contribute to retroflect the particles. Rather, the particles are found to veer within cyclonic eddies or meanders. To clarify this point, we modified Fig. 5 which now shows three examples of trajectories: one where a particle is diverted east by cyclonic features (the most common case, Fig. 5a), one where a particle is blocked by an anti-cyclonic eddy (Fig. 5b), and one where a particle moves westward even in the presence of an anti-cyclonic eddy close to the tip of the Grand Banks (Fig. 5c). We also substantially modified the text to not only discuss the role of cyclonic features in entraining the particles, but also explain the increased presence of anti-cyclonic eddies during high retroflection events and discuss their role in retroflecting the particles. We refer the reviewer to the new version of Section 2.3, at L201-L223.
- On the new version of Fig. 5, we mark the presence of both types of eddies.
- In order to better illustrate the interactions between the particles and the eddy field, we followed the reviewer's recommendation and modified Fig. 5 to show a single particle as it approaches the tip of the Grand Banks. Instead of showing multiple snapshots, we chose to use time labels along the trajectory, on top of the SSH field at the time when the particle reaches the tip of the Grand Banks (when the particle veers east or west). The SSH field does not vary significantly over the days spent by the particle near the tip of the Grand Banks, and hence we do not believe that showing snapshots of SSH at

each time steps would help make our point. The evolution of the SSH field with time is already illustrated in Fig. S6, which shows snapshots of the SSH field through time.

Reviewer: 3. Depending on the answers to the previous question, I might change my view on this comment, but I found that the overall framing of the paper’s main argument presents a false choice between “local” and “remote” control of the degree of retroflection. These two ideas do not seem mutually exclusive. It is plausible that there could be wind-driven changes to the subpolar gyre and the strength of the Labrador Current that allow the Gulf Stream and its eddies to impinge on the bathymetry more often (as suggested by the SSH-retroflection index correlation, which is the highest correlation among all the variables). With greater proximity of the Gulf Stream to the bathymetry, there would be more likelihood for any Labrador Current particles that stray from the bathymetry to be swept to the east. With this in mind, I think it would be better to focus the framing of the paper around the positive result (remote forcing of the Labrador Current retroflection) instead of the negative supposition (local control is secondary), which I was less convinced by given the evidence produced so far. This would mainly alter the Abstract, Section 2.3 and the Discussion.

Authors: We thank the reviewer for raising this point, that was also brought up by Reviewer #3 and helped us reshape the paper. It makes more sense to reconcile the diverse perspectives discussed in the literature, instead of opposing them. We had classified the diverse perspectives into “remote” and “local” forcings, but the results of these studies are not in direct contradiction. Furthermore, we agree that a focus on the large-scale forcing is more consistent with the methodology used here than an in depth analysis of the mechanisms at the tip of the Grand Banks. Hence, we made the following modifications:

- Overall, in the revised manuscript, we no longer oppose “remote” and “local” processes. In section 2.2 (previously called “Remote forcing” now called “Large-scale forcing”), we discuss all forcings, including the SSH (which was previously in the “local” section). Since we argue that the SSH is modulated by an adjustment of circulation under large-scale dynamics, this part logically falls under Section 2.2. In section 2.3 (previously called “Local forcing” now called “Retroflection at the tip of the Grand Banks”), we describe how the particles veer at the tip of the Grand Banks, offering a descriptive view of local processes instead of a mechanistic view that would oppose to the remote, large-scale forcings.
- We modified the title of the paper from “*Remote control of the retroflection of the Labrador Current*” to “*Large-scale control of the retroflection of the Labrador Current*”
- We modified the introduction and the conclusion to avoid opposing “remote” and “local” forcings, and rather focus our findings on large-scale forcing. In the introduction, we replaced “*It has been proposed that the retroflection of the Labrador Current is forced either remotely, upstream of the retroflection point, or locally, at the tip of the Grand Banks.*” by “***Several drivers of the retroflection of the Labrador Current have been proposed in the literature.***”

In the conclusion, we removed the following opening sentence: “*There is no consensus*

yet on whether the retroflexion of the Labrador Current is controlled by remote forcing, by local forcing (i.e. interactions with the Gulf Stream), or by a combination of both (see section 1).” as well as the following sentence: “and considering the absence of a systematic effect of local circulation features at the tip of the Grand Banks on the retroflexion (Fig. 5), we support the hypothesis that the retroflexion is mostly controlled remotely, by wind and the large-scale ocean circulation in the North Atlantic, while the local forcing only plays a secondary role.”.

We also modified the concluding statement as follows: “To conclude, our Lagrangian analysis highlights the major role of remote **large-scale** forcing through winds and gyre dynamics in controlling the retroflexion of the Labrador Current (Fig. 6), **consistent with** results of previous studies that suggested such a link (Jutras et al., 2020; Han et al., 2019; Peterson et al., 2017; New et al., 2021).”, and removed the following sentence: “Local interactions with the Gulf Stream (Neto, 2021; Townsend et al., 2015; Urrego-Blanco and Sheng, 2012) are found to play a secondary role in the retroflexion.”.

- We moved the reference to interactions between the Labrador Current and eddies at the tip of the Grand Banks further down in the discussion, where it occupies a less central position: “**The relation between a strong retroflexion and a northward shift of the Gulf Stream can thus be associated with a large-scale adjustment of the circulation. In fact, whereas anti-cyclonic eddies from the Gulf Stream occasionally block the westward propagation of the Labrador Current waters near the tip of the Grand Banks, they do not appear as a necessary condition for the retroflexion to happen (Section 2.3). Instead, the retroflected waters follow the cyclonic meanders of the Labrador Current.**”
- In the abstract, we removed the sentence “whereas eddies and meanders arising from interactions between the Labrador Current and the Gulf Stream play a secondary role”. This provided space to add details about the results, and we added the following sentence: “[...] **strong retroflexion generally occurs when a large-scale circulation adjustment, related to the subpolar gyre, accelerates the Labrador Current and shifts the Gulf Stream northward, partly driven by a northward shift of the zero-wind-stress-curl line in the western North Atlantic. Starting in 2008, a particularly strong northward shift of the Gulf Stream dominates the other drivers.**”.

Minor comments

Reviewer: Abstract and L27 -It is strange to me to talk about how the Labrador Current carries the water into the subpolar North Atlantic because it originates in the subpolar North Atlantic. I’d say “transports within” rather than “carries into,”

Authors: Thank you for this precision. We applied the reviewer’s suggestion at both locations.

Reviewer: L31 - “originating in the subarctic” be more specific about what you designate as the start of the LC.

Authors: This sentence was modified to “*The Labrador Current is fed by a combination of waters flowing from the West Greenland Current and exported from the Arctic along the Labrador Shelf (Fig. 1).*”

Reviewer: L76 - seesaw implies one high while the other low, which I believe to be true but cannot be seen from Figure 1 trajectories

Authors: The seesaw can be seen on the bottom panel of Fig. S14, which shows the volume transport at different sections along the path of the Labrador Current. The volume transport on the Labrador Shelf (sections SI, WB and BB) is anti-correlated to that on the Scotian Shelf, beyond the tip of the Grand Banks (sections SESP and HL). We added a reference to this figure at L80.

Reviewer: Just a note to say that I find Figure 2 extremely attractive and helpful for understanding

Authors: Thank you.

Reviewer: Everywhere, the Neto et al, 2021 reference should be Gonçalves Neto et al., 2021 (the lead author has 2 last names).

Authors: Thank you for noticing, the reference was corrected in the revised manuscript.

Reviewer: Line 146-150; Does the dependence on wind stress vary by depth? Direct wind forcing would be likely to influence only the shallowest layers, whereas the curl could set up differences in the entire geostrophic transport

Authors: As mentioned in the response to comment #2b, we tested this by computing the time series of the retroreflection index for particles initialized at different depth ranges. We find no major differences between the different depths (Fig. II), suggesting that the mechanisms proposed in this paper are not depth-dependent. We added the following in the Method section: “*The variability of the index is the same for all particles, irrespective of the depth at which they were seeded.*”

Reviewer: 200-202 The lead sentence is repetitive with introduction. The Discussion could simply start with the sentence that is now second.

Authors: This sentence does not appear in the revised manuscript.

Reviewer: 209 - The Solodoch results do not rule out Gulf Stream interactions. Particles from the leaky Labrador Current only get swept far to the east in the presence of the Gulf Stream/NAC. Plus, that paper focuses upstream of the tip

Figure II: Retroflection index, computed from particles initialized in the surface ocean (<100 m) and at depth (> 100 m).

of the Grand Banks.

Authors: We agree with the reviewer that a direct comparison with the results from the Solodoch et al. paper cannot be made here, so we simply removed this sentence.

Reviewer: L342 - I think the pink lines are on Fig 1b (not 1a).

Authors: We thank the reviewer for noticing this error, it is corrected in the revised manuscript.

Reviewer: Figure 1: Why is the pink line longer in 1c than 1b?

Authors: This is to account for the fact that some of the observational platforms drift at deeper depths than the virtual particles, which requires extending the line further offshore, onto the deeper portion of the shelf. This was explained, for the seeding line, in section 4.4, but we modified the text to include all lines:

“All lines extend further offshore than those used for the virtual particles (Fig. 1b,c) to account for the fact that Argo floats typically drift deeper than the virtual particles, hence further offshore on the continental slope”.

Reviewer: L368 - Is this the same technique as Foukal and Lozier 2017 (<https://agupubs.onlinelibrary.wiley.com/doi/10.1029/2017GL074001>) - if so, citation required.

Authors: This method is different, since it uses the 3D velocity field instead of the SSH field. Since we no longer use this index in the revised manuscript, we did not add a citation.

Reviewer: Supplemental references: Jutras, Planat, and Dufour has no reference year. I assume this is the article “in prep.” I am not sure how the journal handles this kind of reference.

Authors: We will make sure to remove this reference if the corresponding manuscript is not accepted for publication by the time this one is published.

Response to Reviewer #3

Comments

Reviewer: The manuscript from Jutras and colleagues highlights the important role of the Labrador Current (LC) in transporting cold, fresh and well oxygenated water not only southwestward along the western boundary but also in the subpolar North Atlantic thanks to an eastward retroflexion. This role is often neglected by the literature that usually focus on the northward transport of warmer and salty water from the subtropical gyre to the subpolar gyre through the Gulf Stream and NAC. In this sense this paper is an important contribution to better understand the role of this water mass in the subpolar region. The paper is well written and I would like to see this paper published, however I have 2 main concerns and some more specific comments that I would like the author to address before the paper is published.

Authors: We thank the reviewer for a very thorough evaluation of our manuscript and for raising important points. We provide point-by-point answers to all the comments below. Throughout the response, bold pieces of text inside citations represent text that was added or modified to the revised version of the manuscript.

Major comments

Reviewer: 1. My first major point regards the use of correlation to explain which forcing are more important in causing the retroflexion. The authors developed an index that gives an information on how weak or strong the retroflexion is. Thus, with negative values the retroflexion is weak (I assume when is equal to -1 means there is no retroflexion at all) and most of the water is exported southwestward along the slope Sea and with positive values the retroflexion is strong (+1 I assume means all the LC turns eastward) and most of the water recirculate in the subpolar gyre moving along the NAC and few is exported southwestward. The authors give some explanations on which forcing are acting on the retroflexion, whether remote or local forcing. My mainly concern is actually on the interpretation of the wind stress as one of the main remote forcing causing this retroflexion. I honestly do not see from what you showed how the wind can play such an important role, and this might be because of how the authors chose to show this dependency. The authors give some correlations (Figure 3) and for example the correlation with the retroflexion index and the wind stress curl is -0.28 which is in my opinion really low. However, the authors says that this is one of the main forcing. On the other hand, the SSH that has much higher correlation (0.62) falls into this local forcing which are argued in the paper are not playing an important role in the retroflexion of the LC. Maybe I miss something in your argument which means it needs to be better explained. I am actually wondering

if these correlations are of any meaning at all and might be misleading. Especially because as far as I understood they are calculated on a time series that refers only to some small subregion which are arbitrary chosen. And about this point I really do not see any meaning to show the wind stress anomaly time series for such a small area when the anomaly (see figure 4 b) looks really patchy. So why those subregions and not other? Does the wind stress anomaly time series in figure 3 have any meaning at all?

Authors: We thank the reviewer for this comment that made us realize that the presentation of our results and of how they were obtained was not clear. In the following, we split our answer into the different points raised by the reviewer.

- The ± 1 values for the retroreflection index do not correspond to whether or not all the water is retroflected or going west, but rather to an anomaly relative to the mean state. The retroflecting branch is always stronger than the westward-flowing branch. The normalization of the index is described in the Method at L367. In the revised manuscript, we clarify the normalization as follows: “The *smoothed* index is then normalized between -1 and 1, and the *mean state* over the whole study period (1993-2015) *is removed*”.
- Our objective was to use the correlations only to provide a visual support to the reader (of what the time evolution looked for some variables), in support of our analyses, which are based on the composite (Fig. 4) and correlation maps (Fig. S4). We understand from reviewer 3’s comment that this was not clear. We definitely agree that correlations are not sufficient to support our hypotheses. Composites and correlation maps are a more useful tool, since they provide spatial information and do not require choosing region to perform averages. In the revised manuscript, we focus on the composite maps, since they contrast the strong and weak events, hence helping us highlight the forcing mechanisms in a complex system. We also modified the manuscript to reduce the use of correlations in our argumentation, given they are not crucial in supporting our conclusions. We refer reviewer #3 to comment #1 by reviewer #2 for a list of these changes. Here are the main ones:
 - One of the reason why the correlations appear to be so important in our reasoning is probably because Fig. 3, which shows these correlations, comes first in the Results section. We modified it to focus on presenting the retroreflection index and its relation to the Labrador Current transport, for which no composite map exists, and SSH anomalies at the tip of the Grand Banks, for which it is interesting to see the shift towards higher values with time. We moved the salinity and temperature anomaly time series (Fig. 3b) to the supplementary material, adding them to Fig. S1. We removed the time series of the wind stress curl in a small region (Fig. 3d), since it was not used, and of the subpolar gyre extent (see next comment).
 - Throughout the text, we now refer to composite maps instead of correlations to support our argumentation. We only discuss correlation coefficients with the NAO, AO and AMOC indices, as these time series cannot be studied with maps. For the volume transport of the Labrador Current, we refer the reader to the new composite

map of the barotropic streamfunction, that we added to Fig. 4. A negative anomaly represents a convergence of the streamlines, hence an acceleration of the current, while a positive anomaly represents a distancing of the streamlines, hence a slow-down of the current. In section 2.2 (Remote forcing), we also moved the discussion of the correlation with the volume transport of the Labrador Current to later in the text. In the discussion, we removed “*Based on correlations found between [...]*”.

- At L90, in the section on the validation of our index, we now refer to composite maps instead of correlations: “*The index **follows** temperature and salinity in the subpolar North Atlantic, in the Slope Sea, and over the northeastern American Shelf [...]*”.
- Reviewer #2 had some suggestions on how to properly compute the correlations coefficients to make sure they were significant, which we applied in the revised manuscript. Following this, we hatched the non-significant correlations zones on Fig. S4.
- About the relative importance of the role of wind forcing and other forcing mechanisms, including SSH, we have two things to note. First, it was not our intention to suggest that the wind was playing a more important role than the other forcing mechanisms, in particular large-scale circulation adjustments. We understand from the reviewer’s comment and from a similar comment by reviewer #1 that this was not clear. Hence, we made the following modifications:
 - We moved the discussion on the SSH higher up in the text, reducing the apparent emphasis on the wind and putting forward the importance of this driver. It is now the first point in section 2.2 (Large-scale forcing), where we directly discuss its relation to the other forcing mechanisms. Since we argue that the SSH and the Gulf Stream position respond to large-scale forcing related to the wind and subpolar gyre dynamics, it makes more sense to include this statement in this section than in section 2.3, which now focuses on the local dynamics of retroflexion at the tip of the Grand Banks.
 - We made a number of modifications throughout the text to ensure that the role of SSH is put forward and that it does not appear like we are suggesting that the wind is playing the most significant role. For example, in the discussion: “***The position of the Gulf Stream, the Labrador Current’s strength and the westerly winds, all of which** are strongly related (Zhang et al., 2016) through the subpolar gyre dynamics (Boning et al., 2006), **are found to play a key role in the retroflexion (Section 2.2).***”;
 in the conclusion: “*To conclude, our Lagrangian analysis highlights the major role of **large-scale** forcing through winds and gyre dynamics in controlling the retroflexion of the Labrador Current (Fig. 6), [...]*”;
 and in the abstract: “*show that **strong retroflexion generally occurs when a large-scale circulation adjustment, related to the subpolar gyre, accelerates the Labrador Current and shifts the Gulf Stream northward, partly driven by a northward shift of the zero-wind-stress-curl line in the western North Atlantic.***”.

Second, reflecting on this comment and on others, it now appears important to discuss some specific periods, which highlight that there is no single driver dominating the whole period. We discuss how the three identified forcings (current’s strength, SSH and winds) align or not in some specific periods. In particular, we discuss the strong shift in the SSH anomalies in 2008 and how it could indicate a predominant role of the Gulf Stream position on the retroflexion in recent years compared to the beginning of the studied period. Here are some of the main additions we brought to the text:

- At L162: *“The strength of the Labrador Current was likely a predominant driver of retroflexion during the 1994-1996 period, when the Gulf Stream was retracted to the south, but the retroflexion was strong (Fig. 3).”*
 - At L181: *“Winds may have played a particularly important role from 1996 to 1998, when the retroflexion was extremely weak, the SSH was at its lowest value over the whole time period (Fig. 3c), but the Labrador Current was strong, in contrast to the rest of the time series (Fig. 3b and supplementary figure S5). During that period, [...]”*
 - At L240: *“Nonetheless, starting in 2008, the proximity of the Gulf Stream to the tip of the Grand Banks may have enhanced the retroflexion by squeezing the Labrador Current meanders that retroreflect the waters. Therefore, the position of the Gulf Stream has likely played a predominant role in driving the retroflexion in recent years (Neto-Goncalvez, 2021; Townsend et al., 2015).”*
- Furthermore, we acknowledge that the correlation between the wind stress curl and the retroflexion index is significant only for a specific region, over the Labrador Shelf and slope. For the other forcings, the relations are significant over a larger area (e.g. the SSH or the SLP). First, Holliday et al. (2020) also noted that the wind stress curl over the Labrador Shelf and slope is strongly linked with the retroflexion. Second, the areas used in computing the time series originally shown in Fig. 3 were based on the largest square region (defined as a box) that showed a strong correlation with the retroflexion index (supplementary figure S4). Whereas the regions might look small on the maps, they are all of more than 100 000 km³, which we believe are not that small. Using boxes to perform averages greatly simplified the computations, compared to using larger regions with more complex definitions. Still, we made the following modifications, to reduce the use of these regions in our argumentation:
 - By removing time series unnecessary to the analysis, that were only used to provide a visual support of the time evolution, we got rid of the region used for the wind stress curl. We only kept the region used to obtain the SSH time series, and moved the salinity and temperature region and time series to the supplementary material.
 - Instead of referring to a weak correlation coefficient, we now refer to the composite map for the wind stress curl (Fig. 4b), on which we can see that weak/strong retroflexion is associated with specific wind conditions. The correlation coefficient over the whole time series (not only focusing on the weak/strong events) can still be found on Fig. S4b. As we explain in the text (Section 3) and as was suggested by

Holliday et al. (2020), the fact that high retroflection is associated with anomalies in a specific region over the Labrador Shelf suggests that the effect of the wind on retroflection works not only through indirect effects like strengthening the current and shifting the large-scale circulation, but also through direct effects, pushing waters offshore at a specific location.

In summary, we have greatly deemphasized the importance of the correlations in the interpretation of the results and shifted the spotlight on the role of the wind stress curl to the SSH.

Reviewer: 2. My second major point regards Fig. 6 which I assumed is the key figure that summarize all the results. When I look at the figure and compared with the one from Holliday et al., (2020), that inspired this schematic, I see something that do not quite fit with figure 10 from Holliday et al., (2020). During strong retroflection the authors show a contracted subpolar gyre, this is exactly the opposites of what is shown and explained in Holliday et al., (2020). During weak retroflection instead the authors show an expanded subpolar gyre which does not fit with the results and conclusions drawn by Holliday et al., (2020). I am also confused about the extension and contraction of the Subpolar gyre (SPG) as explained by the authors. As far as I know a weak gyre is also a contracted gyre and a strong gyre is an expanded gyre (see e.g. Berx & Payne, (2017); Bersch, (2002); Häkkinen & Rhines, (2004); Hátún et al., (2005)) and that is also what it is shown in Holliday et al., 2020. The authors however described exactly the opposite. Comparing figure 6 (upper panel) with figure 10f from Holliday et al., 2020, everything agrees beside the gyre that in figure 10f is expanded while in this manuscript is contracted. The opposite is for the figure 6 (lower panel) compared with figure 10d and e) from Holliday et al., 2020. I assume your conclusions come from what you observed by just looking at the zero-line of wind stress curl in figure 4b and figure S2. These figures show indeed that during strong retroflection period the zero-line of the wind stress curl is shifted northward and during weak retroflection is shifted southward, which would be an indication of a contracted gyre during strong retroflection and an expanded gyre during weak retroflection. However, the figures show only the line until 35°W and not the entire gyre. Could it be that if you have the full domain (the same domain you showed for the Sea level pressure in figure 4d), that would actually show that in the eastern North Atlantic the zero-line shift northward during weak retroflection and southward during strong retroflection in agreement to what Holliday et al., 2020 also found? See for example their figure 9 where the wind stress curl is shown. In 2009 in the western North Atlantic the zero-line is shifted southward but on the eastern side is shifted northward. This year correspond to one of the years where you have a weak retroflection. In 2014 in their figure the zero-line is shifted northward in the western side of the North Atlantic and southward in the eastern side of the NA, this period corresponds to a period of strong retroflection in the manuscript. The fact that there is a northward versus southward shift in the eastern North Atlantic of the wind stress curl opposite to what is observed in the western north Atlantic which influences the position of the subpolar front is not something new. See for example the paper from Bersch (2002) that compares a NAO high period

with a NOW low period. I recommend to include this paper into your discussion since his conclusions are exactly in line (except for this expansion/contraction of the gyre) with your conclusion. Bersch is also talking about the LC retroflexion and how that influence the transport of LC either to the east along the NAC or southward along the boundary. So to conclude my suggestion is to look at the full domain of the wind stress curl (for full domain I mean the entire north Atlantic from west to east) if that is in agreement with Holliday et al., 2020 and Bersch (2002) and if not to discuss this disagreement. I would be surprise however if they disagree.

Authors: We thank the reviewer for the detailed comparison of our and Holliday et al. (2020)'s schematic representation of oceanic and atmospheric forcing, and for proposing an explanation for the observed discrepancies.

- We would first like to clarify that our evaluation of the areal extent of the subpolar gyre was not based on the position of the zero-wind-stress-curl line, but on the area defined by a contour of the barotropic streamfunction over the whole North Atlantic Ocean. The calculation of this “index” was explained in the Method section. This method was different from the ones used in the papers cited by the reviewer, which are based on SSH, and hence a direct comparison is challenging (see next point). We no longer use this index (see below).
- About the relation between the strength and size of the gyre, there is no consensus in the literature - even an ongoing debate - on which index best represents the different aspects of the subpolar gyre dynamics. This debate is highlighted in Foukal and Lozier (2017), noted by the reviewer, but also in Hatun and Chafik (2018) and in Koul et al. (2020), and is related to the discrepancy between the eastern and western North Atlantic Ocean basins, as raised by the reviewer and explored below. Since several studies contradict each other, it is not surprising that our findings contradict some of them. The subpolar gyre is typically defined by the velocity of the water circulating cyclonically around the North Atlantic Ocean basin, and the use of the SSH method has usually been motivated by the fact that velocity measurements over the whole basin do not exist. We chose to work directly from the horizontal water velocity field, given that we use a model and these fields are available.
- We thank the reviewer for bringing the Bersch (2002) reference to our attention. We added the reference to that paper in the revised introduction at L47-48:
*“The retroflexion would be controlled by the wind patterns over the Labrador Shelf (Holliday et al., 2020; Peterson et al., 2017; **Bersch, 2002**) and by the strength of the Labrador Current (Jutras et al., 2020b; Han et al., 2019; **Bersch et al., 2002**; Pickart et al., 1999).”*
as well as at L50:
*“Some studies have suggested that a weak Labrador Current retroflexion is concurrent with a strong North Atlantic Oscillation (NAO, Luo et al., 2006; Pershing et al., 2001), while others have related it to a weak NAO (**Bersch, 2002**), as well as with a strong AMOC (New et al., 2021; Saba et al., 2016).”*
We also included the reference in the discussion at L270: *“To conclude, our Lagrangian*

analysis highlights the major role of **large-scale** forcing through winds and gyre dynamics in controlling the retroflexion of the Labrador Current (Fig. 6), consistent with results of previous studies that suggested such a link (New et al., 2021; Jutras et al., 2020b; Han et al., 2019; Peterson et al., 2017; **Bersch et al., 2002**).”

- About the eastern extent of our analysis, we extended all the composite maps to include the eastern side of the North Atlantic Ocean basin. This new perspective provides interesting results that motivate the use of composites maps of the barotropic streamfunction instead of an estimate of the total extent of the gyre to discuss the subpolar gyre dynamics. First, by extending the zero wind-stress-curl line and the SSH analysis east, we do find opposite states, similar to a dipole, during high and low retroflexion on both sides of the basin (new Fig. 4b,c). Second, if we look at the expansion and contraction of the streamlines of our barotropic streamfunction (that form the subpolar gyre), we find that, while the western side of the gyre contracts during strong retroflexion (as we had previously reported), its eastern side expands (Fig. 4e), as reported by Holliday et al. (2020). This resolves the discrepancy between our Fig. 6 and the Holliday et al. (2020) schematic diagram. In the revised manuscript, we added the composite map of the barotropic streamfunction to Fig. 4 and S3, and removed all references to the time series of the subpolar gyre extent. We also modified Fig. 6 (summary schematic) to highlight the contrasted state of the eastern and western sides of the basin. The discussion of the role of the subpolar gyre, in section 2.2, now reads as follows (L164-L170):

“Both the position of the Gulf Stream/NAC and the Labrador Current’s strength are known to be related to the subpolar dynamics: the NAC forms the southern limb of the gyre, and the Labrador Current forms its western limb. We find that, during strong retroflexion, the subpolar gyre is contracted in the western basin, and expanded in the eastern basin (Fig. 4f). The contraction in the western basin is associated with an acceleration of the Labrador Current as discussed earlier, while the expansion in the eastern basin allows for the propagation of the retroflected water further east (Fig. 4a).”

In the discussion, we specified the side of the basin where the changes occur at L229-L234:

“Strong retroflexion periods coincide with an increased meridional atmospheric pressure gradient in the subpolar North Atlantic, leading to stronger and poleward shifted westerly winds in the western basin. [...] A poleward shifted line of zero wind stress curl in the west of the basin contracts the western portion of the gyre [...].”

- As the reviewer expected, we do find that the zero-wind-stress-curl line is shifted south in the western basin during weak retroflexion, but shifted north in the eastern basin. This is consistent with the above discussion on the contrasted behaviour in both sections of the basin, and with what is seen in Fig. 9 of the Holliday (2020) paper. Since this manuscript focuses on the forcing mechanisms of the retroflexion, we do not explicitly mention the northward shift in the eastern basin, but it is visible on Fig. 4b and implied in the added references mentioned above.

In summary, extending the analysis towards the east allowed us to reconcile our results with

Holliday et al. (2020) about the subpolar gyre dynamics.

Minor comments

Reviewer: Line 80: I am not sure you can cite a paper that is in prep.

Authors: We are hoping that this paper will be accepted by the time this one is. We will make sure to update the citation or to remove it prior to publication.

Reviewer: Line 97: Put the reference to Fig. 4a separate from the references. I thought that you were referring to Fig. 4a in Perez-Brunius et al., 2004.

Authors: Thanks for pointing this out, we changed the reference to “(Fig. 4a and Holliday et al., 2020; Pérez-Brunius et al., 2004; Fischer and Schott, 2002)”.

Reviewer: Line 106: I think the time period when the retroreflection is significantly weak is wrong. It should be 1996-1998. In 1999 it is already positive.

Authors: We modified the statement to “*is particularly weak from 1996 to well into 1998, [...]*”. Note that the marks on the x-axis refer to the beginning of the year. We added a note to that effect in the figure caption.

Reviewer: Line 106: I would also mention the other periods when the reflection is strong, not only the period 2011-2014.

Authors: We changed the text to: “*The retroreflection is particularly weak from 1996 to well into 1998, in 2007, and in 2008-2009, and strong in 1994-1996, 1999, 2002, and 2011-2014. The weak 1996-1998 and strong 2011-2014 periods are particularly salient.*”

Reviewer: Line 107: Can you observe the same freshening also in 1999-2000 and 1993-1995 since the retroreflection index is also positive?

Authors: The composite maps of salinity in Fig. 4 and S3 do include all these events, and show that a stronger retroreflection is concurrent with fresher waters.

Reviewer: Line 112: I am confused, in your time series index the 2009 is negative (it has even green color) so mean weak retroreflection. However, you mentioned that your funding are consistent with float observations which show that more Argo and RAFOS were retroflected in the period 2012-2014 (positive retroreflection index) and 2009 which is negative. So, it is not consistent.

Authors: In our time series (Fig. 3a), the index is strong in 2009. It was weak from mid-2006 to the end of 2008, but becomes strong after that. We specify in the Fig. 2 caption that the tick marks correspond to the beginning of each calendar year, not the middle of the year:

“The tick marks on the time axis indicate the beginning of the year.”

Reviewer: Lines 126 to 129: The authors wrote that “A strong current is generally associated with a strong retroflection as suggested by the positive correlation between the Labrador Current volume transport on the Labrador Shelf and the detrended retroflection index” which is the case for example in 2011-2014. But then in the period in 1996-1998 which is the weakest retroflection should correspond according to this correlation to a weak current. However, in Figure 3b I see the strongest transport on the Labrador Shelf. I am still wondering if these correlations and the way they are calculated are of any meaning at all. Besides, a correlation coefficient of 0.52 might be still considered as low correlated.

Authors: It is true that, from 1996 to 1998, the Labrador Current is strong while the retroflection is weak. The updated correlation coefficients, computed following reviewer #2’s comment, show that the correlation is important only if we do not consider this 1996-1999 period. This is now noted in the revised manuscript:

“Over most of the time series, periods of strong retroflection are associated with periods of strong Labrador Current, as shown by a convergence of the barotropic velocity streamlines in the western part of the subpolar gyre (Fig. 4f), which is indicative of an acceleration of the Labrador Current. A weak but significant positive correlation between the Labrador Current volume transport on the Labrador Shelf and the retroflection index is also found (correlation coefficient of 0.42 for the 1999-2016 period, $p < 0.001$; Fig. 3b and supplementary figure S5.)”

We also added a discussion on this specific period of 1996-1998 a couple of lines below:

“Winds may have played a particularly important role from 1996 to 1998, when the retroflection was extremely weak, the SSH was at its lowest value over the whole time period (Fig. 2c), but the Labrador Current was strong, in contrast to the rest of the time series (Fig. 2b and supplementary figure S5). During that period, the line of zero-wind-stress-curl connected the northwest North Atlantic and Slope Sea region (right panel of supplementary figure S2), which could have urged the waters to move westward despite a strong Labrador Current (see Section 3). A proper evaluation of the drivers of this period would require a more detailed analysis, and could be a subject for future studies.”

Reviewer: Line 132: The authors wrote: “The correlation is negative when considering the volume transport downstream of the Grand Banks”. To be consistent a value should be given, how much negative? However, if you want to follow my suggestions, I think all these correlations might be all removed since I do not see them meaningful.

Authors: This correlation is now given in Fig. 2b of the revised manuscript. As suggested by the reviewer, we now report the correlation coefficient, but we make sure that correlations are not at the center of the discussion.

Reviewer: Line 140-142: The authors wrote “We find a significant anti-correlation

between the retroflection index and the state of the gyre (correlation coefficient of -0.36, $p < 0.0001$)”. Is it significant because of the p value? Because again, -0.36 is not such a strong correlation, they are almost not correlated.

Authors: We agree that this was a weak correlation, and meant that the correlation had statistical significance (small p-value). Nevertheless, following the reviewer’s comment on the definition of the subpolar gyre dynamics, we no longer use the subpolar gyre extent index. In fact, since the contraction/expansion response is different in the eastern and western basins, an estimate of its total area would not provide a proper evaluation of the state of the gyre. We refer the reviewer to our responses to major comment #2 for more details on the impact of these changes on the manuscript.

Reviewer: Lines 140-142: The authors wrote: “Since a contracted (i.e. less extended) gyre is associated with a faster circulation of its peripheral currents, this relation implies that the retroflection is typically higher when the gyre is stronger (faster).” I do disagree with that and I do explain that in my second major comments. The authors should read these papers I have already mentioned above (Berx and Payne 2017, Hátún et al. 2005, Häkkinen and Rhines 2004, Bersch 2002).

Authors: Following the modifications to the text, in response to the reviewer’s main comment on the subpolar gyre dynamics, these sentences were eliminated from the text, and we now mention at L167-L170: “*The contraction in the western basin is associated with an acceleration of the Labrador Current [...]*”, where we refer to the velocity streamfunction. This reasoning is expected from a dynamical perspective, where a convergence of the velocity streamlines implies an acceleration of the current. We no longer mention a direct link between gyre strength and velocity.

Reviewer: Lines 145-147: The authors wrote: “Periods of strong retroflection are associated with negative anomalies in the wind stress curl over the Labrador Shelf and the Grand Banks (Fig. 4b and 3d).” I do not see that from the figures, especially not from figure 3. Besides, the correlation is really low (0.28) as I mentioned in my first major point, my conclusion would be that they are uncorrelated or poorly correlated. Second, in figure 3 the only time when the wind stress curl anomaly is negative is in 2014 the rest of the time series is always positive. Thus, is it only in 2014 that the wind stress curl play a role? I want to stress here again, maybe the way these time series are calculated might be revised. Why did you make the calculation only for such a small region? Why that region and not another one? How different would be the time series if another subregion is chosen? Is it necessary to have a subregion at all? If yes why for each parameter a different subregion is chosen? How meaningful is to compare the times series from different subregions?

Authors: Following the reviewer’s suggestion, we now focus the analysis on the composite maps, and no longer consider the wind stress curl time series (and associated region). From

the composite maps (Fig. 4b and S4), we can see that there is a distinct pattern in the wind stress curl associated with strong and weak retroreflection. This result is consistent with that from Holliday et al. (2020). This association can emerge from the composite map while showing only as a weak correlation because composite maps focus on strong events, and because, as mentioned by the reviewer, the correlation was based on a small region, chosen for convenience. About the sign of the wind stress curl, the reviewer can see that the time series was not centered on zero. Hence, peaks and troughs would be more significant than actual values. This time series is removed from the revised manuscript.

Reviewer: Lines 149-153: The authors wrote: “Conversely, periods of weak retroreflection correspond to positive anomalies in the wind stress curl over the Labrador Shelf (Fig. 4b), and to a southward shift in the line of zero wind-stress-curl. The southward shift connects regions of positive wind stress curl located over the Labrador Sea and the Scotian Shelf (Supplementary figure S2), reducing the offshore push of the winds.” In light of my previous comments I suggest to revise this concept. Figure 4b only shows the zero curl west of 35°W.

Authors: Following the reviewer’s comment, we extended the map to the east of the basin, and we do find that the line of zero-wind-stress-curl has a different behaviour in the eastern North Atlantic Ocean basin. We specify this in the revised manuscript at L174-L177:

In section 2.2: *“These anomalies correspond to stronger zonal winds just north of the Grand Banks that push the water offshore, and to a northward shift of the line of zero wind-stress-curl in that area [...]”* and in the discussion: *“A poleward shifted line of zero wind stress curl in the west of the basin contracts the western portion of the gyre, accelerating the Labrador Current, and shifts the Gulf Stream northward (Peterson et al., 2017)”*

Reviewer: Lines 162-164: Is it $p < 0.0001$? and how much is the correlation with the NAO index?

Authors: Yes, there should be a $<$ sign. This was corrected in the revised manuscript. For the correlation coefficient with the NAO, it is not statistically significant ($p > 0.001$), and the number would have no meaning. We therefore do not report it.

Reviewer: Lines 200-212: I am not sure you can exclude the local forcing and say that the retroreflection is mostly controlled remotely by the wind and the large scale circulation based on the correlations you found in the figure 3. If you base your conclusion on that figure, I would even argue that the local forcing are more important since the highest correlation is between the retroreflection and the SSH.

Authors: This follows comment #3 by reviewer #2. We agree that the position of the Gulf Stream tracked by the SSH anomalies in the Grand Banks region plays a predominant role in the retroreflection, and relate the Gulf Stream north-south migration to an adjustment of the circulation at large scale, tied with the gyre dynamics. As mentioned earlier, we were thinking of local forcing as the direct, causal diversion of the waters by the Gulf Stream meanders

and eddies, and we do not observe that such interactions are a necessary condition for the retroflection to happen (Fig. 5). In the revised manuscript, we clarified this view by better explaining what we mean by a blocking effect (see comment below). We have also reframed the discussion to avoid challenging “Remote” and “Local” effects. Instead, we now focus on the “large-scale” forcing and describe how the retroflection occurs “locally”, at the tip of the Grand Banks. For all the adjustments, see response to comment #3 by reviewer #2. The main modifications are that we moved the results on the SSH to section 2.2 (“Large-scale forcing”). In section 2.3 (now called “Retroflection at the tip of the Grand Banks”), we now focus on describing how the retroflection takes place and how the waters follow or not eddies and meanders. We also increase the emphasis on the role of the presence of the Gulf Stream where it was originally missing, for example in the discussion: “*The position of the Gulf Stream, the Labrador Current’s strength and the westerly winds, all of which are strongly related (Zhang et al., 2016) through the subpolar gyre dynamics (Boning et al., 2016), are found to play a key role in the retroflection (Section 2.2).*”. Not based on Fig. 3 but rather on Fig. 5 and section 2.3, we now expose that the retroflected particles do not appear to be diverted following interactions with anti-cyclonic eddies and meanders shed by the Gulf Stream. Rather, the retroflected particles generally follow cyclonic meanders and eddies that divert them eastward. This suggests that the retroflection is not all due to what we had previously called “local” effects, and that we now rather describe as interactions with warm, Gulf Stream anti-cyclonic eddies. We substantially modified section 2.3 and Fig. 5 to better support this statement (see comment below on the blocking effect).

Reviewer: Lines 207-209: Yes, there is no interaction between the LC and the Gulf Stream at that latitude, but there is an interaction with the LC and the NAC which is the continuation of the Gulf Stream.

Authors: We agree with the reviewer, and we simply removed this sentence. This also addresses a similar comment by reviewer #2 on the same imprecision when we refer to the Solodoch paper, at L209-212, a sentence that we also removed.

Reviewer: Lines 215-248: See my second major comment.

Authors: Following the reviewer’s comment, the manifestation of a contrasted response in the eastern and western North Atlantic basins shows that the Labrador Current accelerates while the gyre expands to the east, maintaining the relevance of these paragraphs. Moreover, in accordance with the suggestion of the reviewer to move away from looking at the correlations, we added some discussion on an early period of the time series, when the SSH was low and the retroflection was high. For that period, the mechanism proposed in these lines is particularly useful to explain the retroflection. We discuss it as follows: In section 2.2, after introducing the correlation with the SSH, we added “*The strength of the Labrador Current was likely a predominant driver of retroflection during the 1994-1996 period, when the Gulf Stream was retracted to the south, but the retroflection was strong (Fig. 3).*” and, at the end paragraph mentioned by the reviewer, “*This mechanism can also explain the discrepancy between SSH and the retroflection index in 1994-1996, when a strong Labrador Current could have yielded a stronger*

retroreflection despite a Gulf Stream positioned far from the Grand Banks.”

Reviewer: Lines 245-248: I still don't understand how can you exclude the blocking effect of the Gulf Stream.

Authors:

- Please see responses to comment #3 by reviewer #2 for modifications that remove the direct opposition between large-scale remote forcings and local forcing. Overall, we modified the manuscript not to oppose but to reconcile the diverse perspectives discussed in the literature.
- In our sense, a blocking effect of the Gulf Stream meant an active, causal blocking, in the sense that the Gulf Stream pushes the Labrador Current away. However, when looking at how the particles retroreflect (Fig. 5), we find that there are only a few occasions when anti-cyclonic eddies generated by the Gulf Stream divert or block the southward flowing particles. Moreover, from 1993 to 1996, the retroreflection is strong but the Gulf Stream is retracted. We now mention this in Section 2.2 at L162-L164:

“The strength of the Labrador Current was likely a predominant driver of retroreflection during the 1994-1996 period, when the Gulf Stream was retracted to the south, but the retroreflection was strong (Fig. 3).” and in the discussion at L263-L265: *“This mechanism can also explain the discrepancy between SSH and the retroreflection index in 1994-1996, when a strong current could have triggered a stronger retroreflection, even in the absence of a blocking effect by the Gulf Stream.”*

This suggests that an actual blocking effect cannot explain all the retroreflection. Since the position of the Gulf Stream is related to large-scale adjustments of the circulation driven by atmospheric and oceanic forcing, the retroreflection would not occur because of a pure active blocking effect by the Gulf Stream, but because it moves north as the retroreflection increases. In other words, we agree with the reviewer that a northward shift of the Gulf Stream comes with an increased retroreflection, and we now say so more explicitly in the revised manuscript, but we wish to say that this is not due to a local effect whereby Gulf Stream waters actively divert the Labrador Current waters. We clarified this vision of a blocking effect (see comment #3 by reviewer #2), for example through the following modifications of the manuscript:

In section 2.3, by reformulating as follows: *“Yet, the trajectories of the particles in that area reveal that the retroflected particles are not systematically blocked or diverted by anti-cyclonic features near the tip of the Grand Banks. Retroflected particles are mostly diverted eastward as they follow cyclonic features of circulation, especially meanders, associated with the tongue of the Labrador Current (Fig. 5a). Anti-cyclonic features sometimes appear to block particles that are then retroflected (Fig. 5b), but particles retroreflect even in the absence (Fig. 5a) and move westward even in the presence (Fig. 5c) of such features..”*

In the discussion, by adding *“The relation between a strong retroreflection and a northward shift of the Gulf Stream can thus be associated with a large-scale adjustment of the circulation. In fact, whereas anti-cyclonic eddies from the Gulf Stream*

occasionally block the westward propagation of the Labrador Current waters near the tip of the Grand Banks, they do not appear as a necessary condition for the retroflection to happen (Section 2.3). Instead, the retroflected waters follow the cyclonic meanders of the Labrador Current.”

Following suggestions by reviewer #2, we also substantially modified Fig. 5 and section 2.3 (see responses to reviewer #2’s main comment #2).

- Nevertheless, we agree with the reviewer that the Gulf Stream could have played a predominant role in driving the retroflection from 2008, when we observe a dramatic northward shift in its position. With the Gulf Stream so close to the tip of the Grand Banks, even if Gulf Stream eddies and meanders do not appear to actively block the southward flowing Labrador Current (Fig. 5), the presence of the Gulf Stream could squeeze the tongue of Labrador Current Waters that appears to divert the particles, fostering retroflection. We add the following to the text: *“Nonetheless, starting in 2008, the proximity of the Gulf Stream to the tip of the Grand Banks may have enhanced the retroflection by squeezing the Labrador Current meanders that retroflect the waters. Therefore, the position of the Gulf Stream has likely played a predominant role in driving the retroflection in recent years (Goncalvez Neto, 2021; Townsend et al., 2015).”*

This significant shift could explain why recent studies suggested an important role for the Gulf Stream in driving the retroflection, a role that does not seem to hold true when considering longer time scales.

Reviewer: Line 251: results instead of resultts

Authors: Thank you, this was corrected in the revised manuscript.

Reviewer: Line 269: It is now called Copernicus marine Service (CMS)

Authors: Thank you, this was corrected in the revised manuscript.

Reviewer: Line 280: AVISO does not provide anymore altimetry-derived surface level anomaly (SLA). This is distributed by Copernicus Marine Service.

Authors: This was corrected in the revised manuscript.

Reviewer: Lines 285-286: Could this overestimation of the WBCs affects the result of the study?

Authors: The cited study looked at WBCs in general. As we mention at L308 and in supplementary material section C, it seems that the volume transport of the Labrador Current is underestimated by up to 30%. This could affect the magnitude of the retroflection, but, since the modelled and observed jets are of the same order of magnitude, we do not think that it would affect the temporal evolution of the retroflection, which is the focus of this study. Accordingly, we added the following sentence to the text:

“Since this study focuses on the temporal variability of the retroflection and not

on its magnitude, this slight underestimation should not affect the results nor our interpretations.”

Reviewer: Lines 291-296: From the comparison it seems like GLORYS is not able to distinguish at all between the Labrador shelf-break and the shelf jet. Is there no other comparison possible with other sections to check this? And if that is the case, how would this affect your results?

Authors: We believe that the reviewer refers to Fig. S11 (now S10) of the supplementary material. In this figure, we see that Glorys is able to distinguish between the two branches of the Labrador Current. In the original version of the figure, the jet in the Glorys output was not very well defined, but this was partially due to the fact that we averaged the Glorys output over all July months, whereas the observations show the average of 10 snapshots. This smoothed the modelled field. In the revised manuscript, we replace the Glorys averages by an average of the 10 days.

For comparison with other sections, we have data at one other line, across Flemish Cap (Fig. III). This figure shows that the main circulation features are captured by the model (strong negative meridional velocities west of the Flemish Pass and close to the shelf-break). This figure also highlights how snapshots of the Glorys data look sharper than averages.

Figure III: Snapshots of the zonal (left), meridional (middle) and total (right) velocity along the Flemish Cap line, in observations (top) and in the Glorys reanalysis (bottom), on Nov. 24, 2013.

Reviewer: Line 310: 1000 meters instead of one kilometer.

Authors: This was corrected in the revised manuscript.

Reviewer: Line 342: “(pink lines on Fig. 1a)”, there are no pink lines on Fig. 1a

Authors: We should have referred to Fig. 1b. This was corrected in the revised manuscript.

Reviewer: Line 364: The authors wrote “The AMOC transport at 26°N is computed by the CMEMS team”. Beside that it is CMS but shouldn’t they be acknowledged?

Authors: CMS was acknowledged in the revised manuscript. The AMOC transport was downloaded from the same website, so we added a mention saying so in the “Data availability” section.

Reviewer: Figure 1: I am a bit confused by figure 1b and 1c. If the blue represents the trajectories for the floats that retroreflect and in green the one that goes southwestward I would not expect any green close to the Flemish Cap and toward the east. But I can see some green at 45°W or even more eastward between 50°N and 40°N. Moreover, in Figure b and c the longitude ticks are missing. In the figure caption you wrote that you consider the shelf and shelf-break branches of the LC together and refer to them as the LC. Is that because you cannot distinguish the two branches with GLORYS?

Authors:

- The green trajectories visible in the east are associated with particles that move westward first before being later entrained towards the east. Particles are tracked for 3 years, and hence can travel a lot of distance through their lifetime. The complete trajectories are shown on Fig. 1. By the time they reach east of the Grand Banks, the waters that had first moved west would have lost their Labrador Current signature, due to prolonged contact with Slope Sea waters. Since our interest in understanding the retroreflection lies mostly in the fact that the retroreflection affects the water properties in the export regions, it appears reasonable, on first order, not to classify these particles as retroflected. We add a note in this in the caption of the figure: “*The green trajectories found in the eastern region correspond to particles that move eastward after having initially moved westward.*”
- The longitude marks are visible at the top of panels b and c.
- We consider the two branches together because we wish to look at the retroreflection of the whole Labrador Current, and not only of one of its branches. We clarify this by reformulating the caption of Fig. 1 as follows: “*In this paper, we **are interested** in both the shelf and shelf-break branches of the Labrador Current. **Hence, we consider them together and refer to them as the Labrador Current.***”

Reviewer: Figure 2: I have problem to understand what the grey bar means. Usually, a bar on a plot is an uncertainty symbol. But I don’t think that is the case.

Authors: We have changed the grey bar for a curly brace (see Fig. IV). The purpose of this brace is to highlight the loss of particles at the tip of the Grand Banks.

Figure IV: New version of Fig. 2 of the revised manuscript.

Reviewer: Figure 3: I would consider to rethink about how the time series are calculated and about the correlations. Moreover, it is really not understandable how the SPGi is calculated and what represents. There are plenty of papers dealing with SPGi, which approach did you use? Which one you follow? This is not at all mentioned in the paper and do not see this particular index useful for your discussion.

Authors: As mentioned above, the SPGi used here was calculated based on the area of a selected value of the barotropic streamfunction, but this index is removed from the revised manuscript. Instead, we discuss the subpolar gyre dynamics based on the composite maps of the barotropic streamfunction.

Reviewer: Figure 4: Why do you only show for a b and c a smaller domain that for figure 4d when the summary in your figure 6 includes the same domain as in figure 4 d? Wouldn't make sense to have all on the same domain, the larger one and see if the zero line of the wind stress curl agrees with Hollidays et al., 2020? I repeat myself, why these subregions? Why a different subregion for each parameter?

Authors: The reasoning for choosing a larger domain for the SLP was that, for this variable, large-scale patterns such as the NAO/AO could contribute to the variability of the retroflection, whereas, for the other variables (SSH, wind), we are not interested in the effects in the eastern basin. In the revised manuscript, we provide a basin-wide view for all variables.

Reviewer: Figure 6: See my second major comment.

Authors: We modified the representation of the subpolar gyre to show the asymmetric modulations of the subpolar gyre under strong and weak retroflection. This new schematic is in agreement with what is reported in the figure of Holliday et al. (2020). We illustrate

the contraction and expansion of the gyre using streamlines to highlight the effect (acceleration/deceleration) on the currents.

REVIEWERS' COMMENTS

Reviewer #1 (Remarks to the Author):

Dear editor and authors,

This is my second review of the manuscript, and I am satisfied that the authors have implemented all my suggestions. I think that comments by the other reviewers have also helped make the manuscript stronger. I listed the merits of this study in my previous review, and I am now confident that the text is ready for publication. I recommend accepting the manuscript in the present form.

Reviewer #2 (Remarks to the Author):

This revised manuscript provides a very comprehensive exploration of the drivers of interannual variability in the recirculation of the Labrador Current from 1993-2015. I applaud the authors for addressing all of the reviewer's comments in a very thorough and meaningful way. The revised text respects the subtleties and ambiguities of this exploration, revealing several driving mechanisms probably acting in concert, and periods where the main hypothesis does not hold up as well as others. For these reasons, I believe the revised manuscript is suitable for publication after attention to a few remaining details.

My only remaining major issue has to do with Figure 5 and its interpretation on lines 215-223. In each of the three anecdotal trajectories shown in Figure 5, the particle is diverted to the east when there is a low SSH feature to its north and high SSH feature to its south. Of course, the SSH gradient between these features provides an eastward geostrophic velocity that can retroflect the particle. It's impossible to say from the anecdotes whether the driver is the formation of a LC cyclonic eddy or the presence of a Gulf Stream anticyclonic eddy. For this reason, I find that the text on 215-223 draws a false choice between the importance of the different-polarity eddies. It's especially strange to dwell on the role of the cyclones given the evidence that the periods of strong retroflection coincide with periods of heightened anticyclonic eddy presence. I wonder if the text on 215-223 is meant to give some context to the excellent paragraph on 245-265 that argues for a key role for the Labrador Current strength? But that later paragraph stands on its own - and nicely explains the potential role of inertia and beta-compensation in a clear and compelling way. With that said, I don't see the purpose of the paragraph on Line 215-223 and suggest removing it.

Related to this concern, Supplemental Figure S13 does not stand on its own. The caption needs additional information for the reader to understand it. What depth are these RAFOS floats traveling at? At what position along the green trajectory is float on the date of the SSH map? What is the time/space resolution of this observational SSH product? These questions are critical to understanding the "no interaction," which I can only make sense of if either:

- a) the RAFOS float is deep enough for the to be advected in a velocity field that is substantially different from the surface flow due to baroclinicity or
- b) for the SSH resolution in space or time to be hiding that this zonally-elongated cyclonic feature is actually two cyclones in the process of merging or splitting and the float passes between them.

Minor issues:

60-61. Simplify sentence structure. Something like the following would be easier to read: Here, we reconcile these different perspectives by presenting evidence that retroflection occurs in a context of large-scale adjustment of the northwestern Atlantic winds and the subpolar gyre circulation, especially the strength of the Labrador Current and position of the Gulf Stream.

134: "Northward shift of GS" is not explicitly shown in Fig 3c - it is just a record of SSH, which indicates a greater presence of the Gulf Stream or its eddies near the GB. The reason to be precise on this is because other studies have gone to great lengths to look at GS position (e.g. Dong et al., 2019; <https://www.nature.com/articles/s41598-019-42820-8> using observations, and Chi et al.

2021 in models <https://agupubs.onlinelibrary.wiley.com/doi/abs/10.1029/2021GL093113>). They cannot define a GS east of $\sim 55^{\circ}\text{W}$, where the GS breaks into a series of eddies and a central axis is not well defined. These studies do not find a northward shift where the GS is well-defined west of 55°W .

144 & 162: I also do not think it's fair to call the 1994-1996 period "anomalously strong retroreflection." Before detrending, the values are 0. The detrending is important to make the correlations stationary, but the raw values better reflect the actual state of the system at any given time. A more basic statement: Though SSH has anomalously low values from 1993-1998, the retroreflection is only anomalously weak for half that period (1996-1998).

Figure 6: What is the gray shading meant to depict?

Reviewer #3 (Remarks to the Author):

Large-scale control of the retroreflection of the Labrador Current by Jutras et al.

I am really satisfied on how the authors addressed all the concerns rose by the reviewers. Well done! It has now a nice discussion, it nicely fit with other studies but still gives some new insight to help to better understand this dynamic region. I do not have any further comments beside a really tiny one: line 489: Fox A.D. et al., it is now actually published. So I recommend to publish the paper as it is.

**Authors' responses to reviewers' comments -
round #2**

Large-scale control of the retroflection of the Labrador Current

Mathilde Jutras, Carolina O. Dufour, Alfonso Mucci, Lauryn L. Talbot

Manuscript Reference Number: NCOMMS-22-36412B

Date: March 26, 2023

Throughout this document, bold text in the author’s response indicates text that was added.

Response to Reviewer #2

Comments

Reviewer: My only remaining major issue has to do with **Figure 5** and its interpretation on lines **215-223**. In each of the three anecdotal trajectories shown in **Figure 5**, the particle is diverted to the east when there is a low SSH feature to its north and high SSH feature to its south. Of course, the SSH gradient between these features provides an eastward geostrophic velocity that can retroreflect the particle. It’s impossible to say from the anecdotes whether the driver is the formation of a LC cyclonic eddy or the presence of a Gulf Stream anticyclonic eddy. For this reason, I find that the text on **215-223** draws a false choice between the importance of the different-polarity eddies. It’s especially strange to dwell on the role of the cyclones given the evidence that the periods of strong retroreflection coincide with periods of heightened anticyclonic eddy presence. I wonder if the text on **215-223** is meant to give some context to the excellent paragraph on **245-265** that argues for a key role for the Labrador Current strength? But that later paragraph stands on its own - and nicely explains the potential role of inertia and beta-compensation in a clear and compelling way. With that said, I don’t see the purpose of the paragraph on Line **215-223** and suggest removing it.

Authors: The objective of this paragraph is to discuss the potential local effect of anti-cyclonic eddies and meanders on the retroreflection of the Labrador Current. Although, as the reviewer mentions, we suggest a mechanism for the retroreflection later in the paper that is not related to these features, we feel like this discussion is required because a number of papers have suggested that anticyclonic eddies and meanders are central in driving the retroreflection of the Labrador Current. In particular, a very recently published paper (Gonçalves-Neto et al., 2023, doi:10.1029/2022JC018756) suggests a major role for anti-cyclonic eddies. This paper was not published at the time we wrote this section. Now that it is, we believe that mentioning it and its results will help to clarify the purpose of this paragraph. We modified the first paragraph of Section 2.3 as follows:

*In recent years, the increased presence of the Gulf Stream at the tip of the Grand Banks concurrent with a strong retroreflection of the Labrador Current has led to the hypothesis that interactions between the Labrador Current and the Gulf Stream could drive the retroreflection (Gonçalves Neto et al., 2023; Gonçalves Neto et al., 2021; Townsend et al., 2015; Urrego-Blanco and Sheng, 2012). **Gulf Stream eddies and meanders have been shown to force the retroreflection of the Labrador Current at the tail of the Grand Banks, in particular since 2008 (Goncalves Neto, 2023).***

The text presented in L215-233 hence provides counter-examples that show that retroreflecting particles do not systematically interact with anti-cyclonic features. In our figure, the

particles do not always retroreflect when there is a high SSH feature to their south. It is for example not the case for the particle in Fig. 5a. Even if there is such a feature near the 01-22 time label, there is none around the 12-26 time label, where the particle detaches from the continental slope. Moreover, in Fig. 5c, there is a high SSH to the south and a low SSH to the north of a particle that does not retroreflect. Hence, while we agree that it is impossible to say what the driving mechanisms are simply by looking at these examples, the counter-examples show that anti-cyclonic features cannot be the only driving mechanism for the retroreflection. This opens the floor for the mechanisms presented in the Discussion section (previously at L245-265).

We rather observe that the particles follow cyclonic features, which explains why we discuss those in the paragraph. Still, we moved this portion of the discussion to the end of the paragraph to avoid any confusion with anti-cyclonic features earlier on.

We modified the second paragraph of Section 2.3 as follows to clarify that the examples serve to contradict the hypothesis of a systematic "blocking effect" and to soften some of the statements:

*Yet, the trajectories of the particles in that area reveal that the retroflected particles are not systematically blocked or diverted by anti-cyclonic features near the tip of the Grand Banks. Retroflected particles are mostly diverted eastward as they follow cyclonic features of circulation, especially meanders, associated with the tongue of the Labrador Current (Fig. 5a). Anti-cyclonic features sometimes appear to block particles that are then retroflected (Fig. 5b), but particles **also** retroreflect even in their absence (Fig. 5a) and move westward even in their presence (Fig. 5c) of such features. Hence, whereas there usually **typically** is an increased presence of Gulf Stream eddies and meanders at the tip of the Grand Banks during strong retroreflection events, these features do not appear to directly contribute to or actively drive **as necessary for** the retroreflection of the Labrador Current to happen. Retroflected particles are rather mostly diverted eastward as they follow cyclonic features of circulation, especially meanders, associated with the tongue of the Labrador Current (Fig. 5a).*

Reviewer: Related to this concern, Supplemental Figure S13 does not stand on its own. The caption needs additional information for the reader to understand it. What depth are these RAFOS floats traveling at? At what position along the green trajectory is float on the date of the SSH map? What is the time/space resolution of this observational SSH product? These questions are critical to understanding the "no interaction," which I can only make sense of if either: a) the RAFOS float is deep enough for the to be advected in a velocity field that is substantially different from the surface flow due to baroclinicity or b) for the SSH resolution in space or time to be hiding that this zonally-elongated cyclonic feature is actually two cyclones in the process of merging or splitting and the float passes between them.

Authors: We agree with the reviewer that more information is needed to properly evaluate potential interactions between the eddies and the floats. In particular, while we can give information on the depth of the floats, we cannot know how deep the eddies reach in the water

Figure I: Updated version of figure S13 (now removed from the manuscript). Caption: Similarly to Fig. 5, examples of situations that show (a) the absence of interaction between a float and an eddy and (b) the deviation of a float by a cold-core cyclonic eddy. Float #1586 drifts at a depth of about 490 m, and float #1579 at a depth of about 530 m. The black arrows point the eddies of interest. Both floats are RAFOS floats. The green line shows the trajectory of the float, and the colour shading indicates the sea level height anomaly at the time when the float reaches the tip of the Grand Banks, indicated in the bottom along with the float number. The black line delineates the 350 m isobath.

column, given that we have no observation in the subsurface concurrent with the passage of the floats. Properly addressing this problem would require extensive analyses that would reach beyond the scope of this study. Hence, we answer some of the reviewer’s questions below, but decided to remove this figure from the supplementary material of the manuscript.

- a) Float #1586 drifts at a depth of about 490 m and float #1579 at a depth of about 530 m. Similar RAFOS floats have detected eddies in that region, suggesting that eddies reach these depths (Bower et al., 2013; Richardson, 1983). However, as we mentioned, we cannot directly confirm their presence due to a lack of data.
- b) We can add time labels similar to those on Fig. 5 to better compare the float position with the date of the SSH contours (Fig. I). This made us realize that there was a mismatch in the timing of the SSH field with the possible ”no interaction” event we point. We correct this by using an SSH snapshot from a different time. This new field also shows that the float ”ignores” the SSH field. Regarding the resolution, we now display the SSH field without interpolation, so as to show the real resolution of the data product.

Since we removed the figure, we also removed the following sentence in the supplementary material:

Moreover, the floats show similar behaviour compared to virtual particles in terms of interactions with eddies and meanders and the tip of the Grand Banks: some floats seem to be deviated by cyclonic eddies, while others are not (Fig. S13).

The figure was not referred to in the main manuscript.

Minor comments

Reviewer: L60-61. Simplify sentence structure. Something like the following would be easier to read: Here, we reconcile these different perspectives by presenting evidence that retroflexion occurs in a context of large-scale adjustment of the northwestern Atlantic winds and the subpolar gyre circulation, especially the strength of the Labrador Current and position of the Gulf Stream.

Authors: As recommended, we modified the sentence to the following: "*Here, we reconcile these different perspectives by presenting evidence that retroflexion occurs in a context of large-scale adjustment of the subpolar gyre circulation, partially driven by winds and affecting the strength of the Labrador Current and position of the Gulf Stream.*"

L134: "Northward shift of GS" is not explicitly shown in Fig 3c - it is just a record of SSH, which indicates a greater presence of the Gulf Stream or its eddies near the GB. The reason to be precise on this is because other studies have gone to great lengths to look at GS position (e.g. Dong et al., 2019; <https://www.nature.com/articles/s41598-019-42820-8> using observations, and Chietal. 2021 in model <https://agupubs.onlinelibrary.wiley.com/doi/abs/10.1029/2021GL093113>). They cannot define a GS east of $\sim 55^{\circ}\text{W}$, where the GS breaks into a series of eddies and a central axis is not well defined. These studies do not find a northward shift where the GS is well-defined west of 55°W .

Authors: We modified "*These positive anomalies are the signature of...*" to "*These positive anomalies suggest...*".

L144 162: I also do not think it's fair to call the 1994-1996 period "anomalously strong retroflexion." Before detrending, the values are 0. The detrending is important to make the correlations stationary, but the raw values better reflect the actual state of the system at any given time. A more basic statement: Though SSH has anomalously low values from 1993-1998, the retroflexion is only anomalously weak for half that period (1996-1998).

Authors: The reviewer is right. We modified the text to the following: "*does not appear to be a necessary condition for the retroflexion to occur, as **retroflexion took place** from 1994 to 1996, when the Gulf Stream was located further south (Fig. 3c).*"

and

"*The strength of the Labrador Current was likely a predominant driver of retroflexion during the 1994-1996 period, when the Gulf Stream was retracted to the south, but retroflexion was not shut down.*"

Reviewer: Figure 6: What is the gray shading meant to depict?

Authors: We now specify in the caption that this shading refers to zones with negative wind stress curl, as indicated by the grey circle arrows. We added the following: "*During strong retroflexion, negative wind stress curl anomalies (**pink zone**) over the Labrador Shelf*"

*reinforce zonal winds [...]. During weak retroflexion, regions of positive wind stress curl anomalies (**grey zones**) connect over the Grand Banks area [...].”*

Response to Reviewer #3

Comments

I am really satisfied on how the authors addressed all the concerns rose by the reviewers. Well done! It has now a nice discussion, it nicely fit with other studies but still gives some new insight to help to better understand this dynamic region. I do not have any further comments beside a really tiny one: line 489: Fox A.D. et al., it is now actually published. So I recommend to publish the paper as it is.

Authors: Thank you, the reference was updated in the revised manuscript.